# A Two-Stage Learning-to-Defer Approach for Multi-Task Learning

Yannis Montreuil [1 2 3 *]   Shu Heng Yeo [1 *]   Axel Carlier [4 5]   Lai Xing Ng [2 5]   Wei Tsang Ooi [1 5]

## Abstract

The Two-Stage Learning-to-Defer (L2D) framework has been extensively studied for classification and, more recently, regression tasks. However, many real-world applications require solving both tasks jointly in a multi-task setting. We introduce a novel Two-Stage L2D framework for multi-task learning that integrates classification and regression through a unified deferral mechanism. Our method leverages a two-stage surrogate loss family, which we prove to be both Bayes-consistent and $(\mathcal{G}, \mathcal{R})$-consistent, ensuring convergence to the Bayes-optimal rejector. We derive explicit consistency bounds tied to the cross-entropy surrogate and the $L_1$-norm of agent-specific costs, and extend minimizability gap analysis to the multi-expert two-stage regime. We also make explicit how shared representation learning—commonly used in multi-task models—affects these consistency guarantees. Experiments on object detection and electronic health record analysis demonstrate the effectiveness of our approach and highlight the limitations of existing L2D methods in multi-task scenarios.

## 1. Introduction

Learning-to-Defer (L2D) integrates predictive models with human experts—or, more broadly, decision-makers—to optimize systems requiring high reliability (Madras et al., 2018). This approach benefits from the scalability of machine learning models and leverages expert knowledge to address complex queries (Hemmer et al., 2021). The Learning-to-Defer approach defers decisions to experts when the

*Equal contribution   [1]School of Computing, National University of Singapore, Singapore   [2]Institute for Infocomm Research, Agency for Science, Technology and Research, Singapore   [3]CNRS@CREATE LTD, 1 Create Way, Singapore   [4]IRIT, Université de Toulouse, CNRS, Toulouse INP, Toulouse, France   [5]IPAL, IRL2955, Singapore. Correspondence to: Yannis Montreuil <yannis.montreuil@u.nus.edu>.

*Proceedings of the $42^{nd}$ International Conference on Machine Learning*, Vancouver, Canada. PMLR 267, 2025. Copyright 2025 by the author(s).

learning-based model has lower confidence than the most confident expert. This deference mechanism enhances safety, which is particularly crucial in high-stakes scenarios (Mozannar & Sontag, 2020; Mozannar et al., 2023; Mao, 2025). For example, in medical diagnostics, the system utilizes patient-acquired data to deliver an initial diagnosis (Johnson et al., 2023; 2016). If the model is sufficiently confident, its diagnosis is accepted; otherwise, the decision is deferred to a medical expert who provides the final diagnosis. Such tasks, which can directly impact human lives, underscore the need to develop reliable systems (Balagurunathan et al., 2021).

Learning-to-Defer has been extensively studied in classification problems (Madras et al., 2018; Verma et al., 2023; Mozannar & Sontag, 2020; Mozannar et al., 2023; Mao et al., 2023a) and, more recently, in regression scenarios (Mao et al., 2024h). However, many modern complex tasks involve both regression and classification components, requiring deferral to be applied to both components simultaneously, as they cannot be treated independently. For instance, in object detection, a model predicts both the class of an object and its location using a regressor, with these outputs being inherently interdependent (Girshick, 2015; Redmon et al., 2016; Buch et al., 2017). In practice, deferring only localization or classification is not meaningful, as decision-makers will treat these two tasks simultaneously. A failure in either component—such as misclassifying the object or inaccurately estimating its position—can undermine the entire problem, emphasizing the importance of coordinated deferral strategies that address both components jointly.

This potential for failure underscores the need for a Learning-to-Defer approach tailored to multi-task problems involving both classification and regression. We propose a novel framework for multi-task environments, incorporating expertise from multiple experts and the predictor-regressor model. We focus our work on the *two-stage scenario*, where the model is already trained offline. This setting is relevant when retraining from scratch the predictor-regressor model is either too costly or not feasible due to diverse constraints such as non-open models (Mao et al., 2023a; 2024h). We approximate the *true deferral loss* using a *surrogate deferral loss* family, based on cross-entropy, and tailored for the two-stage setting, ensuring that the loss effectively approximates the original discontinuous loss function. Our theoretical

analysis establishes that our surrogate loss is both $(\mathcal{G}, \mathcal{R})$-consistent and Bayes-consistent. Furthermore, we study and generalize results on the minimizability gap for deferral loss based on cross-entropy, providing deeper insights into its optimization properties. Our contributions are as follows:

(i) **Novelty**: We introduce two-stage Learning-to-Defer for multi-task learning with multiple experts. Unlike previous L2D methods that focus solely on classification or regression, our approach addresses situations where a sole optimal agent has to be selected to jointly handle both tasks in a unified framework.

(ii) **Theoretical Foundation**: We prove that our surrogate family is both Bayes-consistent and $(\mathcal{G}, \mathcal{R})$-consistent for any cross-entropy-based surrogate. We derive tight consistency bounds that depend on the choice of the surrogate and the $L_1$-norm of the cost, extending minimizability gap analysis to the two-stage, multi-expert setting. Additionally, we establish learning bounds for the *true deferral loss*, showing that generalization improves as agents become more accurate.

(iii) **Empirical Validation**: We evaluate our approach on two challenging tasks. In object detection, our method effectively captures the intrinsic interdependence between classification and regression, overcoming the limitations of existing L2D approaches. In EHR analysis, we show that current L2D methods struggle when agents have varying expertise across classification and regression—whereas our method achieves superior performance.

## 2. Related Work

Learning-to-Defer builds upon the foundational ideas of *Learning with Abstention* (Chow, 2003; Bartlett & Wegkamp, 2008; Cortes et al., 2016; Geifman & El-Yaniv, 2017; Ramaswamy et al., 2018; Cao et al., 2022; Mao et al., 2024a), where a model is permitted to abstain from making a prediction when its confidence is low. The core insight of L2D is to extend this framework from rejection to deferral—delegating uncertain decisions to external agents or experts whose confidence may exceed that of the model.

**One-stage Learning-to-Defer.** L2D was originally introduced by Madras et al. (2018) for binary classification, using a *pass function* inspired by the *predictor-rejector* framework of Cortes et al. (2016). In the multiclass setting, Mozannar & Sontag (2020) proposed a *score-based* formulation that leverages a log-softmax surrogate to ensure Bayes-consistency. This formulation has since been extended to a wide range of classification tasks (Okati et al., 2021; Verma et al., 2023; Cao et al., 2024; 2022; Keswani et al., 2021; Kerrigan et al., 2021; Hemmer et al., 2022; Benz & Rodriguez, 2022; Tailor et al., 2024; Liu et al., 2024; Palomba et al.,

2024; Wei et al., 2024). A pivotal contribution by Mozannar et al. (2023) challenged the sufficiency of Bayes-consistency, showing that existing score-based methods may be suboptimal under realizable distributions—particularly when the hypothesis class is restricted. They introduced the notion of *hypothesis-consistency*, which strengthens theoretical alignment between the surrogate loss and the constrained hypothesis space. This work sparked a broader effort to refine the theoretical foundations of L2D using tools from surrogate risk analysis (Long & Servedio, 2013; Zhang & Agarwal, 2020; Awasthi et al., 2022; Mao et al., 2023b). Recent theoretical advances have solidified the status of score-based L2D. Mao et al. (2024f) established that the general score-based L2D framework achieves $\mathcal{H}$-consistency, while Mao et al. (2024g; 2025b) introduced a novel surrogate loss that guarantees *realizable-consistency*—i.e., optimality under realizable distributions. Montreuil et al. (2025c) generalize L2D to deferral to the set of top-$k$ experts. Beyond classification, the L2D framework has also been extended to regression (Mao et al., 2024h), demonstrating its applicability in continuous-output settings with expert deferral.

**Two-stage Learning-to-Defer.** The emergence of large-scale pretrained models has motivated the development of *two-stage* L2D frameworks, where both the model and the expert agents are trained offline. This reflects practical constraints: most users lack the computational resources to fine-tune large models end-to-end. Narasimhan et al. (2022) were the first to formalize this setting, and Mao et al. (2023a) introduced a dedicated *predictor–rejector* architecture tailored for two-stage L2D, with theoretical guarantees including both Bayes- and hypothesis-consistency. Charusaie et al. (2022) offered a comparative analysis of one-stage (joint training) and two-stage (post hoc) L2D, highlighting trade-offs between model flexibility and sample efficiency. More recently, two-stage L2D has been successfully extended to regression (Mao et al., 2024h) and top-$k$ expert deferral (Montreuil et al., 2025b), and has been applied to real-world tasks such as extractive question answering (Montreuil et al., 2024) and adversarial robustness (Montreuil et al., 2025a).

Despite significant progress, current two-stage L2D research largely addresses classification and regression independently. However, many contemporary tasks involve both regression and classification components, necessitating their joint optimization. In this work, we extend two-stage L2D to joint classifier-regressor models, addressing this critical gap.

## 3. Preliminaries

**Multi-task scenario.** We consider a multi-task setting encompassing both classification and regression problems. Let $\mathcal{X}$ denote the input space, $\mathcal{Y} = \{1, \dots, n\}$ represent the

set of $n$ distinct classes, and $\mathcal{T} \subseteq \mathbb{R}$ denote the space of real-valued targets for regression. For compactness, each data point is represented as a triplet $z = (x, y, t) \in \mathcal{Z}$, where $\mathcal{Z} = \mathcal{X} \times \mathcal{Y} \times \mathcal{T}$. We assume the data is sampled independently and identically distributed (i.i.d.) from a distribution $\mathcal{D}$ over $\mathcal{Z}$ (Girshick, 2015; Redmon et al., 2016; Carion et al., 2020).

We define a *backbone* $w \in \mathcal{W}$, or shared feature extractor, such that $w : \mathcal{X} \to \mathcal{Q}$. For example, $w$ can be a deep network that takes an input $x \in \mathcal{X}$ and produces a latent feature vector $q = w(x) \in \mathcal{Q}$. Next, we define a *classifier* $h \in \mathcal{H}$, representing all possible classification heads operating on $\mathcal{Q}$. Formally, $h$ is a score function defined as $h : \mathcal{Q} \times \mathcal{Y} \to \mathbb{R}$, where the predicted class is $h(q) = \arg\max_{y \in \mathcal{Y}} h(q, y)$. Likewise, we define a *regressor* $f \in \mathcal{F}$, representing all regression heads, where $f : \mathcal{Q} \to \mathcal{T}$. These components are combined into a single multi-head network $g \in \mathcal{G}$, where $\mathcal{G} = \{ g : g(x) = (h \circ w(x), f \circ w(x)) \mid w \in \mathcal{W}, h \in \mathcal{H}, f \in \mathcal{F} \}$. Hence, $g$ jointly produces classification and regression outputs, $h(q)$ and $f(q)$, from the same latent representation $q = w(x)$.

**Consistency in classification.** In the classification setting, the goal is to identify a classifier $h \in \mathcal{H}$ in the specific case where $w(x) = x$, such that $h(x) = \arg\max_{y \in \mathcal{Y}} h(x, y)$. This classifier should minimize the true error $\mathcal{E}_{\ell_{01}}(h)$, defined as $\mathcal{E}_{\ell_{01}}(h) = \mathbb{E}_{(x,y)} \big[ \ell_{01}(h(x), y) \big]$. The Bayes-optimal error is given by $\mathcal{E}_{\ell_{01}}^B(\mathcal{H}) = \inf_{h \in \mathcal{H}} \mathcal{E}_{\ell_{01}}(h)$. However, directly minimizing $\mathcal{E}_{\ell_{01}}(h)$ is challenging due to the non-differentiability of the *true multiclass* 0-1 loss (Zhang, 2002; Steinwart, 2007; Awasthi et al., 2022; Cortes et al., 2025; Mao et al., 2025c). This motivates the introduction of the cross-entropy *multiclass surrogate* family, denoted by $\Phi_{01}^\nu : \mathcal{H} \times \mathcal{X} \times \mathcal{Y} \to \mathbb{R}^+$, which provides a convex upper bound to the *true multiclass loss* $\ell_{01}$. This family is parameterized by $\nu \geq 0$ and encompasses standard surrogate functions widely adopted in the community such as the MAE (Ghosh et al., 2017) or the log-softmax (Mohri et al., 2012).

$$\Phi_{01}^\nu = \begin{cases} \frac{1}{1-\nu}\left( \left[ \sum_{y' \in \mathcal{Y}} e^{h(x,y')-h(x,y)} \right]^{1-\nu} - 1 \right) & \nu \neq 1 \\ \log\left( \sum_{y' \in \mathcal{Y}} e^{h(x,y')-h(x,y)} \right) & \nu = 1. \end{cases}$$
(1)

The corresponding surrogate error is defined as $\mathcal{E}_{\Phi_{01}^\nu}(h) = \mathbb{E}_{(x,y)} \big[ \Phi_{01}^\nu(h(x), y) \big]$, with its optimal value given by $\mathcal{E}_{\Phi_{01}^\nu}^*(\mathcal{H}) = \inf_{h \in \mathcal{H}} \mathcal{E}_{\Phi_{01}^\nu}(h)$. A crucial property of a surrogate loss is *Bayes-consistency*, which guarantees that minimizing the surrogate generalization error also minimizes the true generalization error (Zhang, 2002; Steinwart, 2007; Bartlett et al., 2006; Tewari & Bartlett, 2007; Mao et al., 2025a; Zhong, 2025). Formally, $\Phi_{01}^\nu$ is Bayes-consistent with respect to $\ell_{01}$ if, for any sequence $\{h_k\}_{k \in \mathbb{N}} \subset \mathcal{H}$, the following implication holds:

$$\mathcal{E}_{\Phi_{01}^\nu}(h_k) - \mathcal{E}_{\Phi_{01}^\nu}^*(\mathcal{H}) \xrightarrow{k \to \infty} 0$$
$$\implies \mathcal{E}_{\ell_{01}}(h_k) - \mathcal{E}_{\ell_{01}}^B(\mathcal{H}) \xrightarrow{k \to \infty} 0.$$
(2)

This property assumes that $\mathcal{H} = \mathcal{H}_{\text{all}}$, a condition that does not necessarily hold for restricted hypothesis classes such as $\mathcal{H}_{\text{lin}}$ or $\mathcal{H}_{\text{ReLU}}$ (Long & Servedio, 2013; Awasthi et al., 2022). To address this limitation, Awasthi et al. (2022) proposed $\mathcal{H}$-consistency bounds. These bounds depend on a non-decreasing function $\Gamma : \mathbb{R}^+ \to \mathbb{R}^+$ and are expressed as:

$$\mathcal{E}_{\Phi_{01}^\nu}(h) - \mathcal{E}_{\Phi_{01}^\nu}^*(\mathcal{H}) + \mathcal{U}_{\Phi_{01}^\nu}(\mathcal{H}) \geq$$
$$\Gamma\Big( \mathcal{E}_{\ell_{01}}(h) - \mathcal{E}_{\ell_{01}}^B(\mathcal{H}) + \mathcal{U}_{\ell_{01}}(\mathcal{H}) \Big),$$
(3)

where the minimizability gap $\mathcal{U}_{\ell_{01}}(\mathcal{H})$ measures the disparity between the best-in-class generalization error and the expected pointwise minimum error: $\mathcal{U}_{\ell_{01}}(\mathcal{H}) = \mathcal{E}_{\ell_{01}}^B(\mathcal{H}) - \mathbb{E}_x \big[ \inf_{h \in \mathcal{H}} \mathbb{E}_{y|x} [\ell_{01}(h(x), y)] \big]$. Notably, the minimizability gap vanishes when $\mathcal{H} = \mathcal{H}_{\text{all}}$ (Steinwart, 2007; Awasthi et al., 2022; Cortes et al., 2024; Mao et al., 2024e;d;j;b). In the asymptotic limit, inequality (3) guarantees the recovery of Bayes-consistency, aligning with the condition in (2).

## 4. Two-stage Multi-Task L2D: Theoretical Analysis

### 4.1. Formulating the True Deferral Loss

We extend the two-stage predictor–rejector framework, originally proposed by (Narasimhan et al., 2022; Mao et al., 2023a), to the multi-task setting described in Section 3. Specifically, we consider an *offline-trained model* $g \in \mathcal{G}$, which jointly performs classification and regression. In addition, we assume access to $J$ offline-trained experts, denoted $\mathrm{M}_j$ for $j \in \{1, \ldots, J\}$. Each expert outputs predictions of the form $m_j(x) = \big( m_j^h(x), m_j^f(x) \big)$, where $m_j^h(x) \in \mathcal{Y}$ and $m_j^f(x) \in \mathcal{T}$ correspond to the classification and regression components, respectively. Each expert prediction lies in a corresponding space $\mathcal{M}_j$, so that $m_j(x) \in \mathcal{M}_j$. We denote the aggregated outputs of all experts as $m(x) = \big( m_1(x), \ldots, m_J(x) \big) \in \mathcal{M} := \prod_{j=1}^J \mathcal{M}_j$. We write $[J] := \{1, \ldots, J\}$ to denote the index set of experts, and define the set of all agents as $\mathcal{A} := \{0\} \cup [J]$, where agent 0 corresponds to the model $g$. Thus, the system contains $|\mathcal{A}| = J + 1$ agents in total.

To allocate each decision, we introduce a *rejector* function $r \in \mathcal{R}$, where $r : \mathcal{X} \times \mathcal{A} \to \mathbb{R}$. Given an input $x \in \mathcal{X}$, the rejector selects the agent $j \in \mathcal{A}$ that maximizes its score: $r(x) = \arg\max_{j \in \mathcal{A}} r(x, j)$. This mechanism induces the *deferral loss*, a mapping $\ell_{\text{def}} : \mathcal{R} \times \mathcal{G} \times \mathcal{Z} \times \mathcal{M} \to \mathbb{R}_+$, which quantifies the cost of allocating a decision to a particular agent.

**Definition 4.1** (True deferral loss). Let an input $x \in \mathcal{X}$, for any $r \in \mathcal{R}$, we have the *true deferral loss*:

$$\ell_{\text{def}}(r, g, m, z) = \sum_{j=0}^{J} c_j(g(x), m_j(x), z) 1_{r(x)=j},$$

with a bounded cost $c_j$ that quantifies the penalty incurred when allocating the decision to agent $j \in \mathcal{A}$. When the rejector $r \in \mathcal{R}$ predicts $r(x) = 0$, the decision is assigned to the multi-task model $g$, incurring a base cost $c_0$ defined as $c_0(g(x), z) = \rho(g(x), z)$, where $\rho(\cdot, \cdot) \in \mathbb{R}_+$ measures the discrepancy between the model's output $g(x)$ and the ground truth $z$. Conversely, if the rejector selects $r(x) = j$ for some $j > 0$, the decision is deferred to expert $j$, yielding a deferral cost $c_j(m_j(x), z) = \rho(m_j(x), z) + \beta_j$. Here, $\beta_j \geq 0$ denotes the querying cost associated with invoking expert $j$, which may reflect domain-specific constraints such as computational overhead, annotation effort, or time expenditure.

When the classification and regression objectives are separable, the total cost can be decomposed as $c_j = \lambda^{\text{cla}} c^{\text{cla}} + \lambda^{\text{reg}} c^{\text{reg}}$, where $\lambda^{\text{cla}}, \lambda^{\text{reg}} \geq 0$ specify the relative importance of each task. A neutral setting is recovered when $\lambda^{\text{cla}} = \lambda^{\text{reg}} = 1$, ensuring a task-agnostic trade-off. If classification performance is prioritized, one can select $\lambda^{\text{cla}} > \lambda^{\text{reg}}$ to favor agents with stronger classification expertise.

**Optimal deferral rule.** In Definition 4.1, we introduced the *true deferral loss* $\ell_{\text{def}}$, which quantifies the expected cost incurred when allocating predictions across the model and experts. Our goal is to minimize this loss by identifying the Bayes-optimal rejector $r \in \mathcal{R}$ that minimizes the true risk. To formalize this objective, we analyze the *pointwise Bayes rejector* $r^B(x)$, which minimizes the conditional risk $\mathcal{C}_{\ell_{\text{def}}}$. The corresponding population risk is given by $\mathcal{E}_{\ell_{\text{def}}}(g, r) = \mathbb{E}_x[\mathcal{C}_{\ell_{\text{def}}}(g, r, x)]$. The following lemma characterizes the optimal decision rule at each input $x \in \mathcal{X}$.

**Lemma 4.2** (Pointwise Bayes Rejector). *Given an input $x \in \mathcal{X}$ and data distribution $\mathcal{D}$, the rejection rule that minimizes the conditional risk $\mathcal{C}_{\ell_{\text{def}}}$ associated with the true deferral loss $\ell_{\text{def}}$ is:*

$$r^B(x) = \begin{cases} 0 & \text{if } \inf_{g \in \mathcal{G}} \mathbb{E}_{y,t|x}[c_0] \leq \min_{j \in [J]} \mathbb{E}_{y,t|x}[c_j] \\ j & \text{otherwise,} \end{cases}$$

The proof is provided in Appendix B. Lemma 4.2 shows that the optimal rejector $r \in \mathcal{R}$ assigns the decision to the model $g \in \mathcal{G}$ whenever its expected cost is lower than that of any expert. Otherwise, the rejector defers to the expert with the minimal expected deferral cost.

Although Lemma 4.2 characterizes the Bayes-optimal policy under the true deferral loss $\ell_{\text{def}}$, this loss is non-differentiable and thus intractable for direct optimization in practice (Zhang, 2002).

### 4.2. Surrogate Loss for Two-Stage Multi-Task L2D

**Introducing the surrogate.** To address the optimization challenges posed by discontinuous losses (Berkson, 1944; Cortes & Vapnik, 1995), we introduce a family of convex surrogate losses with favorable analytical properties. Specifically, we adopt the multiclass cross-entropy surrogates $\Phi_{01}^\nu : \mathcal{R} \times \mathcal{X} \times \mathcal{A} \to \mathbb{R}_+$, which upper-bounds the true multiclass 0-1 loss $\ell_{01}$ and facilitates gradient-based optimization. This surrogate family is defined in Equation 1.

Building on the framework of Mao et al. (2024h), who proposed convex surrogates for deferral settings, we extend their approach to account for the interdependence between classification and regression tasks. In our setting, this yields a family of surrogate losses $\Phi_{\text{def}}^\nu : \mathcal{R} \times \mathcal{G} \times \mathcal{M} \times \mathcal{Z} \to \mathbb{R}_+$, which incorporate the full structure of the multi-task cost.

**Lemma 4.3** (Surrogate Deferral Surrogates). *Let $x \in \mathcal{X}$ and let $\Phi_{01}^\nu$ be a multiclass surrogate loss. Then the surrogate deferral loss $\Phi_{def}^\nu$ for $J + 1$ agents is given by*

$$\Phi_{def}^\nu(r, g, m, z) = \sum_{j=0}^{J} \tau_j(g(x), m(x), z) \, \Phi_{01}^\nu(r, x, j),$$

*where the aggregated cost weights are defined as $\tau_j(g(x), m(x), z) = \sum_{i=0}^{J} c_i(g(x), m_i(x), z) 1_{i \neq j}$.*

The surrogate deferral loss $\Phi_{\text{def}}^\nu$ combines the individual surrogate losses $\Phi_{01}^\nu(r, x, j)$ for each agent $j \in \mathcal{A}$, weighted by the corresponding aggregated cost $\tau_j$. Intuitively, $\tau_0$ quantifies the total cost of deferring to any expert instead of using the model, while $\tau_j$ for $j > 0$ reflects the total cost incurred by selecting expert $j$ instead of any other agent, including the model and other experts.

This construction preserves task generality and only requires that the base surrogate $\Phi_{01}^\nu$ admit an $\mathcal{R}$-consistency bound. The modular formulation of the cost functions $c_j$ allows this surrogate to flexibly accommodate diverse multi-task settings.

**Consistency of the surrogate losses.** In Lemma 4.3, we established that the proposed surrogate losses form a convex upper bound on the *true deferral loss* $\ell_{\text{def}}$. However, it remains to determine whether this surrogate family provides a reliable approximation of the true loss in terms of optimal decision-making. In particular, it is not immediate that the pointwise minimizer of the surrogate loss, $r^*(x)$, aligns with the Bayes-optimal rejector $r^B(x)$ that minimizes $\ell_{\text{def}}$. To address this, we study the relationship between

the surrogate and true risks by analyzing their respective *excess risks*. Specifically, we compare the surrogate excess risk, $\mathcal{E}_{\Phi^\nu_{def}}(g, r) - \mathcal{E}^*_{\Phi^\nu_{def}}(\mathcal{G}, \mathcal{R})$, to the true excess risk, $\mathcal{E}_{\ell_{def}}(g, r) - \mathcal{E}^B_{\ell_{def}}(\mathcal{G}, \mathcal{R})$. Understanding this discrepancy is crucial for establishing the $(\mathcal{G}, \mathcal{R})$-consistency of the surrogate loss family, a topic extensively studied in prior work on multiclass surrogate theory (Steinwart, 2007; Zhang, 2002; Bartlett et al., 2006; Awasthi et al., 2022).

Leveraging consistency bounds developed in (Awasthi et al., 2022; Mao et al., 2024c), we present Theorem 4.4, which proves that the surrogate deferral loss family $\Phi^\nu_{def}$ is indeed $(\mathcal{G}, \mathcal{R})$-consistent.

**Theorem 4.4** (($\mathcal{G}, \mathcal{R}$)-consistency bounds). *Let $g \in \mathcal{G}$ be a multi-task model. Suppose there exists a non-decreasing function $\Gamma^\nu : \mathbb{R}_+ \to \mathbb{R}_+$, parameterized by $\nu \geq 0$, such that the $\mathcal{R}$-consistency bound holds for any distribution $\mathcal{D}$:*

$$\mathcal{E}_{\Phi^\nu_{01}}(r) - \mathcal{E}^*_{\Phi^\nu_{01}}(\mathcal{R}) + \mathcal{U}_{\Phi^\nu_{01}}(\mathcal{R}) \geq$$
$$\Gamma^\nu\left(\mathcal{E}_{\ell_{01}}(r) - \mathcal{E}^B_{\ell_{01}}(\mathcal{R}) + \mathcal{U}_{\ell_{01}}(\mathcal{R})\right),$$

*then for any $(g, r) \in \mathcal{G} \times \mathcal{R}$, any distribution $\mathcal{D}$, and any $x \in \mathcal{X}$,*

$$\mathcal{E}_{\ell_{def}}(g, r) - \mathcal{E}^B_{\ell_{def}}(\mathcal{G}, \mathcal{R}) + \mathcal{U}_{\ell_{def}}(\mathcal{G}, \mathcal{R}) \leq$$
$$\overline{\Gamma}^\nu\left(\mathcal{E}_{\Phi^\nu_{def}}(r) - \mathcal{E}^*_{\Phi^\nu_{def}}(\mathcal{R}) + \mathcal{U}_{\Phi^\nu_{def}}(\mathcal{R})\right)$$
$$+ \mathcal{E}_{c_0}(g) - \mathcal{E}^B_{c_0}(\mathcal{G}) + \mathcal{U}_{c_0}(\mathcal{G}),$$

*where the expected aggregated cost vector is given by $\overline{\tau} = \left(\mathbb{E}_{y,t|x}[\tau_0], \ldots, \mathbb{E}_{y,t|x}[\tau_J]\right)$, and $\overline{\Gamma}^\nu(u) = \|\overline{\tau}\|_1 \Gamma^\nu\left(\frac{u}{\|\overline{\tau}\|_1}\right)$ with $\Gamma^\nu(u) = \mathcal{T}^{-1,\nu}(u)$. In the case of the log-softmax surrogate ($\nu = 1$), the transformation is given by $\mathcal{T}^{\nu=1}(u) = \frac{1+u}{2}\log(1 + u) + \frac{1-u}{2}\log(1 - u)$.*

The proof of Theorem 4.4, along with generalizations to any $\nu \geq 0$, is provided in Appendix C. This result yields refined consistency guarantees for the surrogate deferral loss, improving upon the bounds established by Mao et al. (2024h). The bounds are explicitly tailored to the cross-entropy surrogate family and parameterized by $\nu$, allowing for precise control over the surrogate's approximation behavior. Crucially, the tightness of the bound depends on the aggregated deferral costs, and is scaled by the $L_1$-norm $\|\overline{\tau}\|_1$, which quantifies the cumulative cost discrepancy across agents.

Moreover, we show that the surrogate deferral losses are $(\mathcal{G}, \mathcal{R})$-consistent whenever the underlying multiclass surrogate family $\Phi^\nu_{01}$ is $\mathcal{R}$-consistent. Under the assumption that $\mathcal{R} = \mathcal{R}_{all}$ and $\mathcal{G} = \mathcal{G}_{all}$, the minimizability gaps vanish, as established by Steinwart (2007). As a result, minimizing the *surrogate deferral excess risk* while accounting for the minimizability gap yields $\mathcal{E}_{\Phi^\nu_{def}}(r_k) - \mathcal{E}^*_{\Phi^\nu_{def}}(\mathcal{R}_{all}) + \mathcal{U}_{\Phi^\nu_{def}}(\mathcal{R}_{all}) \xrightarrow{k \to \infty} 0$. Since the multi-task model $g$ is trained

offline, it is reasonable to assume that the $c_0$-excess risk also vanishes: $\mathcal{E}_{c_0}(g_k) - \mathcal{E}^B_{c_0}(\mathcal{G}_{all}) + \mathcal{U}_{c_0}(\mathcal{G}_{all}) \xrightarrow{k \to \infty} 0$. Combining the two convergence results and invoking the properties of $\overline{\Gamma}^\nu$, we conclude that

$$\mathcal{E}_{\ell_{def}}(g, r_k) - \mathcal{E}^B_{\ell_{def}}(\mathcal{G}_{all}, \mathcal{R}_{all}) + \mathcal{U}_{\ell_{def}}(\mathcal{G}_{all}, \mathcal{R}_{all}) \xrightarrow{k \to \infty} 0.$$

Hence, the following corollary holds:

**Corollary 4.5** (Bayes-consistency of the deferral surrogate losses). *Under the conditions of Theorem 4.4, and assuming $(\mathcal{G}, \mathcal{R}) = (\mathcal{G}_{all}, \mathcal{R}_{all})$ and $\mathcal{E}_{c_0}(g_k) - \mathcal{E}^B_{c_0}(\mathcal{G}_{all}) \xrightarrow{k \to \infty} 0$, the surrogate deferral loss family $\Phi^\nu_{def}$ is Bayes-consistent with respect to the true deferral loss $\ell_{def}$. Specifically, minimizing the surrogate deferral excess risk ensures convergence of the true deferral excess risk. Formally, for sequences $\{r_k\}_{k \in \mathbb{N}} \subset \mathcal{R}$ and $\{g_k\}_{k \in \mathbb{N}} \subset \mathcal{G}$, we have:*

$$\mathcal{E}_{\Phi^\nu_{def}}(r_k) - \mathcal{E}^*_{\Phi^\nu_{def}}(\mathcal{R}_{all}) \xrightarrow{k \to \infty} 0$$
$$\implies \mathcal{E}_{\ell_{def}}(g_k, r_k) - \mathcal{E}^B_{\ell_{def}}(\mathcal{G}_{all}, \mathcal{R}_{all}) \xrightarrow{k \to \infty} 0.$$

This result confirms that, as $k \to \infty$, the surrogate losses $\Phi^\nu_{def}$ attain asymptotic Bayes optimality for both the rejector $r$ and the offline-trained multi-task model $g$. Thus, the surrogate family faithfully approximates the true deferral loss in the limit. Moreover, the pointwise surrogate-optimal rejector $r^*(x)$ converges to a close approximation of the Bayes-optimal rejector $r^B(x)$, thereby inducing deferral decisions consistent with the characterization in Lemma 4.2 (Bartlett et al., 2006).

**Analysis of the minimizability gap.** In As shown by Awasthi et al. (2022), the minimizability gap does not vanish in general. Understanding the conditions under which it arises, quantifying its magnitude, and identifying effective mitigation strategies are crucial for ensuring that surrogate-based optimization aligns with the true task-specific objectives.

We provide a novel and strong characterization of the minimizability gap in the two-stage setting with multiple experts, extending the results of Mao et al. (2024i), who analyzed the gap in the context of learning with abstention (constant cost) for a single expert and a specific distribution.

**Theorem 4.6** (Characterization of Minimizability Gaps). *Assume $\mathcal{R}$ is symmetric and complete. Then, for the cross-entropy multiclass surrogates $\Phi^\nu_{01}$ and any distribution $\mathcal{D}$, the following holds for $\nu \geq 0$:*

$$\mathcal{C}^{\nu,*}_{\Phi^\nu_{def}}(\mathcal{R}, x) = \begin{cases} \|\overline{\tau}\|_1 H\left(\frac{\overline{\tau}}{\|\overline{\tau}\|_1}\right) & \text{for } \nu = 1 \\ \|\overline{\tau}\|_1 - \|\overline{\tau}\|_\infty & \nu = 2 \\ \frac{1}{\nu-1}\left[\|\overline{\tau}\|_1 - \|\overline{\tau}\|_{\frac{1}{2-\nu}}\right] & \nu \in (1, 2) \\ \frac{1}{1-\nu}\left[\left(\sum_{k=0}^J \overline{\tau}_k^{\frac{1}{2-\nu}}\right)^{2-\nu} - \|\overline{\tau}\|_1\right] & \nu > 2, \end{cases}$$

where $\overline{\tau} = \{\mathbb{E}_{y,t|x}[\overline{\tau}_0], \ldots, \mathbb{E}_{y,t|x}[\overline{\tau}_J]\}$, the aggregated costs are $\tau_j = \sum_{k=0}^{J} c_k \mathbb{1}_{k \neq j}$, and $H$ denotes the Shannon entropy. The minimizability gap is defined as $\mathcal{U}_{\Phi_{def}^\nu}(\mathcal{R}) = \mathcal{E}_{\Phi_{def}^\nu}^*(\mathcal{R}) - \mathbb{E}_x \left[ \mathcal{C}_{\Phi_{def}^\nu}^{\nu,*}(\mathcal{R}, x) \right]$.

We provide the proof in Appendix D. Theorem 4.6 characterizes the minimizability gap $\mathcal{U}_{\Phi_{def}^\nu}(\mathcal{R})$ for cross-entropy multiclass surrogates over symmetric and complete hypothesis sets $\mathcal{R}$. The gap depends on $\nu \geq 0$, and its behavior varies across different surrogates. Specifically, for $\nu = 1$, the gap is proportional to the Shannon entropy of the normalized expected cost vector $\frac{\overline{\tau}}{\|\overline{\tau}\|_1}$, which increases with entropy, reflecting higher uncertainty in the misclassification distribution. For $\nu = 2$, the gap simplifies to the difference between the $L_1$-norm and $L_\infty$-norm of $\overline{\tau}$, where a smaller gap indicates concentrated misclassifications, thus reducing uncertainty. For $\nu \in (1, 2)$, the gap balances the entropy-based sensitivity at $\nu = 1$ and the margin-based sensitivity at $\nu = 2$. As $\nu \to 1^+$, the gap emphasizes agents with higher misclassification counts, while as $\nu \to 2^-$, it shifts towards aggregate misclassification counts. For $\nu < 1$, where $p = \frac{1}{2-\nu} \in (0, 1)$, the gap becomes more sensitive to misclassification distribution, increasing when errors are dispersed. For $\nu > 2$, with $p < 0$, reciprocal weighting reduces sensitivity to dominant errors, potentially decreasing the gap but at the risk of underemphasizing critical misclassifications.

In the special case of learning with abstention and a single expert ($J = 1$), assigning costs $\tau_0 = 1$ and $\tau_J = 1 - c$ recovers the minimizability gap introduced by Mao et al. (2024i). Thus, our formulation generalizes the minimizability gap to settings with multiple experts, non-constant costs, and arbitrary distributions $\mathcal{D}$.

### 4.3. Encoder–Aware Bounds

In this section, we show that our approach is theoretically aligned with multi-task learning using shared representations. Let $\mathcal{W}$ denote a class of representation functions (encoders), $\mathcal{H}$ a class of classification heads, and $\mathcal{F}$ a class of regression heads. For any $(w, h, f) \in \mathcal{W} \times \mathcal{H} \times \mathcal{F}$, the multi-task predictor is defined as $g_{w,h,f}(x) = (h \circ w(x), f \circ w(x))$, where the shared representation $w(x)$ is passed to both task-specific heads. The true risk defined as $\mathcal{E}_{c_0}(g) = \mathbb{E}_{z \sim \mathcal{D}}[c_0(g(x), (y, t))]$, where $z = (x, y, t) \in \mathcal{X} \times \mathcal{Y} \times \mathcal{T}$. The Bayes risk over a class $\mathcal{G}$ is given by $\mathcal{E}_{c_0}^B(\mathcal{G}) = \inf_{g \in \mathcal{G}} \mathcal{E}_{c_0}(g)$.

**Proposition 4.7** (Head and representation gaps.)**.** *Fix $w \in \mathcal{W}$ and let $\mathcal{G} := \{g_{w',h',f'} : w' \in \mathcal{W}, h' \in \mathcal{H}, f' \in \mathcal{F}\}$.*

Define

$$\mathcal{E}_{\min}(w) := \inf_{h',f'} \mathcal{E}_{c_0}\left(g_{w,h',f'}\right),$$

$$\Delta_{\text{heads}}(w, h, f) := \mathcal{E}_{c_0}\left(g_{w,h,f}\right) - \mathcal{E}_{\min}(w),$$

$$\Delta_{\text{repr}}(w) := \mathcal{E}_{\min}(w) - \mathcal{E}_{c_0}^B(\mathcal{G}).$$

The quantity $\Delta_{\text{heads}}$ measures *head sub–optimality* given the extracted representation from the encoder fixed at a particular iteration, while $\Delta_{\text{repr}}$ captures how far $w$ lies from a Bayes–optimal shared representation.

**Lemma 4.8** (Non–negativity of the gaps)**.** *For all $(w, h, f) \in \mathcal{W} \times \mathcal{H} \times \mathcal{F}$, we have $\Delta_{\text{heads}}(w, h, f) \geq 0$. For every $w \in \mathcal{W}$, we have $\Delta_{\text{repr}}(w) \geq 0$.*

*Proof.* Fix $w$. By definition of the infimum, $\mathcal{E}_{\min}(w) = \inf_{h',f'} \mathcal{E}_{c_0}\left(g_{w,h',f'}\right) \leq \mathcal{E}_{c_0}\left(g_{w,h,f}\right)$ for any heads $(h, f)$, hence $\Delta_{\text{heads}}(w, h, f) \geq 0$. For the representation gap, note that $\mathcal{E}_{c_0}^B(\mathcal{G}) = \inf_{w',h',f'} \mathcal{E}_{c_0}\left(g_{w',h',f'}\right) \leq \mathcal{E}_{\min}(w)$, so $\Delta_{\text{repr}}(w) = \mathcal{E}_{\min}(w) - \mathcal{E}_{c_0}^B(\mathcal{G}) \geq 0$. Both inequalities hold with equality when $(w, h, f)$ is Bayes–optimal. $\square$

**Proposition 4.9** (Excess–risk decomposition)**.** *For every $(w, h, f)$,*

$$\mathcal{E}_{c_0}\left(g_{w,h,f}\right) - \mathcal{E}_{c_0}^B(\mathcal{G}) = \Delta_{\text{heads}}(w, h, f) + \Delta_{\text{repr}}(w).$$

*Proof.* Add and subtract $\mathcal{E}_{\min}(w)$. $\square$

**Proposition 4.10** (Cost of Enforcing a Shared Encoder)**.** *Suppose two* independent *heads act directly on the raw input $x$: $g_{sep,h,f}(x) = (h(x), f(x))$. Let $\mathcal{E}_{sep,c_0}^B := \inf_{h,f} \mathcal{E}_{c_0}\left(g_{sep,h,f}\right)$ and define*

$$\Delta_{\text{MTL}} := \mathcal{E}_{c_0}^B(\mathcal{G}) - \mathcal{E}_{sep,c_0}^B.$$

Hence $\Delta_{\text{MTL}} < 0$ indicates that forcing a *shared* encoding is beneficial, whereas $\Delta_{\text{MTL}} > 0$ points to a potential penalty relative to two stand–alone models.

Combining definitions,

$$\mathcal{E}_{c_0}(g_{w,h,f}) - \mathcal{E}_{c_0}^B(\mathcal{G}) = \left[\mathcal{E}_{c_0}(g_{w,h,f}) - \mathcal{E}_{sep,c_0}^B\right] - \Delta_{\text{MTL}},$$

we can link these relationships with the main Theorem 4.4 that states that for any $(g, r) \in \mathcal{G} \times \mathcal{R}$. Setting $g = g_{w,h,f}$ and invoking Proposition 4.9 yields the *encoder–aware consistency bound*:

**Corollary 4.11** (Encoder–aware $(\mathcal{G}, \mathcal{R})$–consistency)**.** *For any $(w, h, f, r) \in \mathcal{W} \times \mathcal{H} \times \mathcal{F} \times \mathcal{R}$,*

$$\mathcal{E}_{\ell_{\text{def}}}(g_{w,h,f}, r) - \mathcal{E}_{\ell_{\text{def}}}^B(\mathcal{G}, \mathcal{R}) + \mathcal{U}_{\ell_{\text{def}}}(\mathcal{G}, \mathcal{R}) \leq$$
$$\overline{\Gamma}^\nu\left(\mathcal{E}_{\Phi_{\text{def}}^\nu}(r) - \mathcal{E}_{\Phi_{\text{def}}^\nu}^B(\mathcal{R}) + \mathcal{U}_{\Phi_{\text{def}}^\nu}(\mathcal{R})\right)$$
$$+ \Delta_{\text{heads}}(w, h, f) + \Delta_{\text{repr}}(w) + \mathcal{U}_{c_0}(\mathcal{G}).$$

Corollary (4.11) decomposes the end–to–end excess deferral risk into three *orthogonal* sources: (i) the rejector optimisation error, (ii) the head sub–optimality, and (iii) the representation gap.

The bound suggests a two–stage pipeline: (i) learn or select high–capacity representations to minimise $\Delta_{\text{repr}}$ as well as best heads for this representation, then (ii) optimise the rejector to tighten the remaining terms. The pipeline exactly mirrors our proposed L2D solution. Such decoupling is particularly attractive when $|\mathcal{T}|$ is large and feature sharing is essential for sample efficiency.

### 4.4. Generalization Bound

We aim to quantify the generalization capability of our system, considering both the complexity of the hypothesis space and the quality of the participating agents. To this end, we define the empirical optimal rejector $\widehat{r}^B$ as the minimizer of the empirical generalization error:

$$\widehat{r}^B = \arg\min_{r \in \mathcal{R}} \frac{1}{K} \sum_{k=1}^{K} \ell_{\text{def}}(g, m, r, z_k), \qquad (4)$$

where $\ell_{\text{def}}$ denotes the *true deferral loss* function. To characterize the system's generalization ability, we utilize the Rademacher complexity, which measures the expressive richness of a hypothesis class by evaluating its capacity to fit random noise (Bartlett & Mendelson, 2003; Mohri et al., 2012). The proof of Lemma 4.12 is provided in Appendix E.

**Lemma 4.12.** *Let $\mathcal{L}_1$ be a family of functions mapping $\mathcal{X}$ to $[0, 1]$, and let $\mathcal{L}_2$ be a family of functions mapping $\mathcal{X}$ to $\{0, 1\}$. Define $\mathcal{L} = \{l_1 l_2 : l_1 \in \mathcal{L}_1, l_2 \in \mathcal{L}_2\}$. Then, the empirical Rademacher complexity of $\mathcal{L}$ for any sample $S$ of size $K$ is bounded by:*

$$\widehat{\mathfrak{R}}_S(\mathcal{L}) \le \widehat{\mathfrak{R}}_S(\mathcal{L}_1) + \widehat{\mathfrak{R}}_S(\mathcal{L}_2).$$

For simplicity, we assume costs $c_0(g(x), z) = \ell_{01}(h(x), y) + \ell_{\text{reg}}(f(x), t)$ and $c_{j>0}(m_j(x), z) = c_0(m_j(x), z)$. We assume the regression loss $\ell_{\text{reg}}$ to be non-negative, bounded by $L$, and Lipschitz. Furthermore, we assume that $m_{k,j}^h$ is drawn from the conditional distribution of the random variable $M_j^h$ given parameters $\{X = x_k, Y = y_k\}$, and that $m_{k,j}^f$ is drawn from the conditional distribution of $M_j^f$ given $\{X = x_k, T = t_k\}$. We define the family of deferral loss functions as $\mathcal{L}_{\text{def}} = \{\ell_{\text{def}} : \mathcal{G} \times \mathcal{R} \times \mathcal{M} \times \mathcal{Z} \to \mathbb{R}^+\}$. Under these assumptions, we derive the generalization bounds for the binary setting as follows:

**Theorem 4.13** (Learning bounds of the deferral loss). *For any expert $M_j$, any distribution $\mathcal{D}$ over $\mathcal{Z}$, we have with*

*probability $1 - \delta$ for $\delta \in [0, 1/2]$, that the following bound holds at the optimum:*

$$\mathcal{E}_{\ell_{def}}(h, f, r) \le \widehat{\mathcal{E}}_{\ell_{def}}(h, f, r) + 2\mathfrak{R}_K(\mathcal{L}_{def}) + \sqrt{\frac{\log 1/\delta}{2K}},$$

*with*

$$\mathfrak{R}_K(\mathcal{L}_{def}) \le \frac{1}{2}\mathfrak{R}_K(\mathcal{H}) + \mathfrak{R}_K(\mathcal{F}) + \sum_{j=1}^{J} \Omega(m_j^h, y)$$
$$+ \Big( \sum_{j=1}^{J} \max \ell_{reg}(m_j^f, t) + 2 \Big) \mathfrak{R}_K(\mathcal{R}),$$

*with $\Omega(m_j^h, y) = \frac{1}{2}\mathcal{D}(m_j^h \ne y) \exp\left(-\frac{K}{8}\mathcal{D}(m_j^h \ne y)\right) + \mathcal{R}_{K\mathcal{D}(m_j^h \ne y)/2}(\mathcal{R}).$*

We prove Theorem 4.13 in Appendix F. The terms $\mathfrak{R}_K(\mathcal{H})$ and $\mathfrak{R}_K(\mathcal{F})$ denote the Rademacher complexities of the hypothesis class $\mathcal{H}$ and function class $\mathcal{F}$, respectively, indicating that the generalization bounds depend on the complexity of the pre-trained model. The term $\Omega(m_j^h, y)$ captures the impact of each expert's classification error on the learning bound. It includes an exponentially decaying factor, $\frac{\mathcal{D}(m_j^h \ne y)}{2} \exp\left(-\frac{K\mathcal{D}(m_j^h \ne y)}{8}\right)$, which decreases rapidly as the sample size $K$ grows or as the expert's error rate $\mathcal{D}(m_j^h \ne y)$ declines (Mozannar & Sontag, 2020). This reflects the intuition that more accurate experts contribute less to the bound, improving overall generalization. Finally, the last term suggests that the generalization properties of our *true deferral loss* depend on the expert's regression performance.

## 5. Experiments

In this section, we present the performance improvements achieved by the proposed Learning-to-Defer surrogate in a multi-task context. Specifically, we demonstrate that our approach excels in object detection, a task where classification and regression components are inherently intertwined and cannot be delegated to separate agents, and where existing L2D methods encounter significant limitations. Furthermore, we evaluate our approach on an Electronic Health Record task, jointly predicting mortality (classification) and length of stay (regression), comparing our results with Mao et al. (2023a; 2024h).

For each experiment, we report the mean and standard deviation across four independent trials to account for variability in the results. All training and evaluation were conducted on an NVIDIA H100 GPU. We give our training algorithm in Appendix A. Additional figures and details are provided in Appendix G. To ensure reproducibility, we have made our implementation publicly available.

## 5.1. Object Detection Task

We evaluate our approach using the Pascal VOC dataset (Everingham et al., 2010), a multi-object detection benchmark. This is the first time such a multi-task problem has been explored within the L2D framework as previous L2D approaches require the classification and regression component to be independent (Mao et al., 2023a; 2024h).

**Dataset and Metrics:** The PASCAL Visual Object Classes (VOC) dataset (Everingham et al., 2010) serves as a widely recognized benchmark in computer vision for evaluating object detection models. It consists of annotated images spanning 20 object categories, showcasing diverse scenes with varying scales, occlusions, and lighting conditions. To assess object detection performance, we report the mean Average Precision (mAP), a standard metric in the field. Additionally, in the context of L2D, we report the allocation metric (All.), which represents the ratio of allocated queries per agent.

**Agents setting:** We trained three distinct Faster R-CNN models (Ren et al., 2016) to serve as our agents, differentiated by their computational complexities. The smallest, characterized by GFLOPS $= 12.2$, represents our model $g \in \mathcal{G}$ with $\mathcal{G} = \{ g : g(x) = (h \circ w(x), f \circ w(x)) \mid w \in \mathcal{W}, h \in \mathcal{H}, f \in \mathcal{F} \}$. The medium-sized, denoted as Expert 1, has a computational cost of GFLOPS $= 134.4$, while the largest, Expert 2, operates at GFLOPS $= 280.3$. To account for the difference in complexity between Experts 1 and 2, we define the ratio $R_G = 280.3/134.4$ and set the query cost for Expert 1 as $\beta_1 = \beta_2/R_G$. This parameterization reflects the relative computational costs of querying experts. We define the agent costs as $c_0(g(x), z) = \text{mAP}(g(x), z)$ and $c_{j \in [J]}(m_j(x), z) = \text{mAP}(m_j(x), z)$. We report the performance metrics of the agents alongside additional training details in Appendix G.1.

**Rejector:** The rejector is trained using a smaller version of the Faster R-CNN model (Ren et al., 2016). Training is performed for 200 epochs using the Adam optimizer (Kingma & Ba, 2017) with a learning rate of 0.001 and a batch size of 64. The checkpoint achieving the lowest empirical risk on the validation set is selected for evaluation.

**Results:** In Figure 1, we observe that for lower cost values, specifically when $\beta_1 < 0.15$, the system consistently avoids selecting Expert 1. This outcome arises because the cost difference between $\beta_1$ and $\beta_2$ is negligible, making it more advantageous to defer to Expert 2 (the most accurate expert), where the modest cost increase is offset by superior outcomes. When $\beta_2 = 0.15$, however, it becomes optimal to defer to both experts and model at the same time. In particular, there exist instances $x \in \mathcal{X}$ where both Expert 1

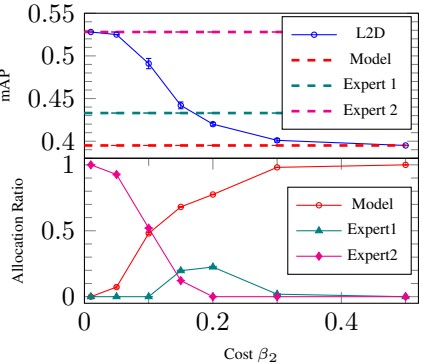

*Figure 1.* Performance comparison across different cost values $\beta_2$ on Pascal VOC (Everingham et al., 2010). The table reports the mean Average Precision (mAP) and the allocation ratio for the model and two experts with mean and variance. We report these results in Appendix Table 3.

and Expert 2 correctly predict the target (while the model does not). In such cases, Expert 1 is preferred due to its lower cost $\beta_1 < \beta_2$. Conversely, for instances $x \in \mathcal{X}$ where Expert 2 is accurate and Expert 1 (along with the model) is incorrect, the system continues to select Expert 2, as $\beta_2$ remains relatively low. For $\beta_2 \geq 0.2$, the increasing cost differential between the experts shifts the balance in favor of Expert 1, enabling the system to achieve strong performance while minimizing overall costs.

This demonstrates that our approach effectively allocates queries among agents, thereby enhancing the overall performance of the system, even when the classification and regression tasks are interdependent.

## 5.2. EHR Task

We compare our novel approach against existing two-stage L2D methods (Mao et al., 2023a; 2024h). Unlike the first experiment on object detection (Subsection 5.1), where classification and regression tasks are interdependent, this evaluation focuses on a second scenario where the two tasks can be treated independently.

**Dataset and Metrics:** The Medical Information Mart for Intensive Care IV (MIMIC-IV) dataset (Johnson et al., 2023) is a comprehensive collection of de-identified health-related data patients admitted to critical care units. For our analysis, we focus on two tasks: mortality prediction and length-of-stay prediction, corresponding to classification and regression tasks, respectively. To evaluate performance, we report accuracy (Acc) for the mortality prediction task, which quantifies classification performance, and Smooth L1 loss (sL1) for the length-of-stay prediction task, which measures the deviation between the predicted and actual values. Additionally, we report the allocation metric (All.) for L2D to

capture query allocation behavior.

**Agents setting:** We consider two experts, $M_1$ and $M_2$, acting as specialized agents, aligning with the category allocation described in (Mozannar & Sontag, 2020; Verma et al., 2023; Verma & Nalisnick, 2022; Cao et al., 2024). The dataset is partitioned into $Z = 6$ clusters using the $K$-means algorithm (Lloyd, 1982), where $Z$ is selected via the Elbow method (Thorndike, 1953). The clusters are denoted as $\{C_1, C_2, \ldots, C_Z\}$. Each cluster represents a subset of data instances grouped by feature similarity, enabling features-specific specialization by the experts. The experts are assumed to specialize in distinct subsets of clusters based on the task. For classification, $M_1$ correctly predicts the outcomes for clusters $C_{\text{cla}}^{M_1} = \{C_1, C_2, C_4\}$, while $M_2$ handles clusters $C_{\text{cla}}^{M_2} = \{C_1, C_5, C_6\}$. Notably, cluster $C_1$ is shared between the two experts, reflecting practical scenarios where domain knowledge overlaps. For regression tasks, $M_1$ is accurate on clusters $C_{\text{reg}}^{M_1} = \{C_1, C_3, C_5\}$, while $M_2$ specializes in clusters $C_{\text{reg}}^{M_2} = \{C_1, C_4, C_6\}$. Here too, overlap is modeled, with cluster $C_1$ being common to both experts and classification-regression task. Note that the category assignments do not follow any specific rule.

We assume that each expert produces correct predictions for the clusters they are assigned (Verma et al., 2023; Mozannar & Sontag, 2020). Conversely, for clusters outside their expertise, predictions are assumed to be incorrect. In such cases, for length-of-stay predictions, the outcomes are modeled using a uniform probability distribution to reflect uncertainty. The detailed performance evaluation of these agents is provided in Appendix G.2.

The model utilizes two compact transformer architectures (Vaswani et al., 2017) for addressing both classification and regression tasks, formally defined as $\mathcal{G} = \{g : g(x) = (h(x), f(x)) \mid h \in \mathcal{H}, f \in \mathcal{F}\}$. The agent's costs are specified as $c_0(g(x), z) = \lambda^{\text{cla}} \ell_{01}(h(x), y) + \lambda^{\text{reg}} \ell_{\text{reg}}(f(x), t)$ and $c_{j \in [J]}(m_j(x), z) = c_0(m_j(x), z) + \beta_j$. Consistent with prior works (Mozannar & Sontag, 2020; Verma et al., 2023; Mao et al., 2023a; 2024h), we set $\beta_j = 0$.

**Rejectors:** The two-stage L2D rejectors are trained using a small transformer model (Vaswani et al., 2017) as the encoder, following the approach outlined by Yang et al. (2023), with a classification head for query allocation. Training is performed over 100 epochs with a learning rate of 0.003, a warm-up period of 0.1, a cosine learning rate scheduler, the Adam optimizer (Kingma & Ba, 2017), and a batch size of 1024 for all baselines. The checkpoint with the lowest empirical risk on the validation set is selected for evaluation.

**Results:** Table 1 compares the performance of our proposed Learning-to-Defer (Ours) approach with two existing methods: a classification-focused rejector (Mao et al., 2023a) and a regression-focused rejector (Mao et al., 2024h). The results highlight the limitations of task-specific rejectors and the advantages of our balanced approach.

| Rejector | Acc (%) | sL1 | All. Model | All. Expert 1 | All. Expert 2 |
|---|---|---|---|---|---|
| Mao et al. (2023a) | $71.3 \pm .1$ | $1.45 \pm .03$ | $.60 \pm .02$ | $.01 \pm .01$ | $.39 \pm .02$ |
| Mao et al. (2024h) | $50.7 \pm .8$ | $1.18 \pm .05$ | $.38 \pm .01$ | $.37 \pm .02$ | $.25 \pm .01$ |
| Ours | $70.0 \pm .5$ | $1.28 \pm .02$ | $.66 \pm .01$ | $.12 \pm .02$ | $.22 \pm .01$ |

*Table 1.* Performance comparison of different two-stage L2D. The table reports accuracy (Acc), smooth L1 loss (sL1), and allocation rates (All.) to the model and experts with mean and variance.

The classification-focused rejector achieves the highest classification accuracy at 71.3% but struggles with regression, as reflected by its high smooth L1 loss of 1.45. On the other hand, the regression-focused rejector achieves the best regression performance with an sL1 loss of 1.18 but performs poorly in classification with an accuracy of 50.7%. In contrast, our method balances performance across tasks, achieving a classification accuracy of 70.0% and an sL1 loss of 1.28. Moreover, it significantly reduces reliance on experts, allocating 66% of queries to the model compared to 60% for Mao et al. (2023a) and 38% for Mao et al. (2024h). Expert involvement is minimized, with only 12% and 22% of queries allocated to Experts 1 and 2, respectively.

Since the experts possess distinct knowledge for the two tasks ($C_{\text{cla}}^{M_1}$ and $C_{\text{reg}}^{M_1}$ for $M_1$), independently deferring classification and regression may lead to suboptimal performance. In contrast, our approach models deferral decisions dependently, considering the interplay between the two components to achieve better overall results.

## 6. Conclusion

We introduced a Two-Stage Learning-to-Defer framework for multi-task problems, extending existing approaches to jointly handle classification and regression. We proposed a two-stage surrogate loss family that is both $(\mathcal{G}, \mathcal{R})$-consistent and Bayes-consistent for any cross-entropy-based surrogate. Additionally, we derived tight consistency bounds linked to cross-entropy losses and the $L_1$-norm of aggregated costs. We further established novel minimizability gap for the two-stage setting, generalizing prior results to Learning-to-Defer with multiple experts. Finally, we showed that our learning bounds improve with a richer hypothesis space and more confident experts.

We validated our framework on two challenging tasks: (i) object detection, where classification and regression are inherently interdependent—beyond the scope of existing L2D methods; and (ii) electronic health record analysis, where we demonstrated that current L2D approaches can be suboptimal even when classification and regression tasks are independent.

## Acknowledgment

This research is supported by the National Research Foundation, Singapore under its AI Singapore Programme (AISG Award No: AISG2-PhD-2023-01-041-J) and by A*STAR, and is part of the programme DesCartes which is supported by the National Research Foundation, Prime Minister's Office, Singapore under its Campus for Research Excellence and Technological Enterprise (CREATE) programme.

## Impact Statement

This paper advances the theoretical and practical understanding of machine learning, contributing to the development of more effective models and methods. While our research does not present any immediate or significant ethical concerns, we recognize the potential for indirect societal impacts.

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

# A. Algorithm

---

**Algorithm 1** Two-Stage Learning-to-Defer for Multi-Task Learning Algorithm

---

**Input:** Dataset $\{(x_k, y_k, t_k)\}_{k=1}^K$, multi-task model $g \in \mathcal{G}$, experts $m \in \mathcal{M}$, rejector $r \in \mathcal{R}$, number of epochs EPOCH, batch size $B$, learning rate $\eta$.
**Initialization:** Initialize rejector parameters $\theta$.
**for** $i = 1$ to EPOCH **do**
    Shuffle dataset $\{(x_k, y_k, t_k)\}_{k=1}^K$.
    **for** each mini-batch $\mathcal{B} \subset \{(x_k, y_k, t_k)\}_{k=1}^K$ of size $B$ **do**
        Extract input-output pairs $z = (x, y, t) \in \mathcal{B}$.
        Query model $g(x)$ and experts $m(x)$.                {Agents are pre-trained and fixed}
        Evaluate costs $c_0(g(x), z)$ and $c_{j>0}(m(x), z)$.           {Compute task-specific costs}
        Compute rejector prediction $r(x) = \arg\max_{j \in \mathcal{A}} r(x, j)$.           {Rejector decision}
        Compute surrogate deferral empirical risk $\widehat{\mathcal{E}}_{\Phi_{\mathrm{def}}}$:
            $\widehat{\mathcal{E}}_{\Phi_{\mathrm{def}}} = \frac{1}{B} \sum_{z \in \mathcal{B}} \left[ \Phi_{\mathrm{def}}(g, r, m, z) \right].$           {Empirical risk computation}
        Update parameters $\theta$ using gradient descent:
            $\theta \leftarrow \theta - \eta \nabla_\theta \widehat{\mathcal{E}}_{\Phi_{\mathrm{def}}}.$                  {Parameter update}
    **end for**
**end for**
**Return:** trained rejector model $r^*$.

---

We will prove key lemmas and theorems stated in our main paper.

# B. Proof of Lemma 4.2

We aim to prove Lemma 4.2, which establishes the optimal deferral decision by minimizing the conditional risk.

By definition, the Bayes-optimal rejector $r^B(x)$ minimizes the conditional risk $\mathcal{C}_{\ell_{\mathrm{def}}}$, given by:

$$\mathcal{C}_{\ell_{\mathrm{def}}}(g, r, x) = \mathbb{E}_{y,t|x}[\ell_{\mathrm{def}}(g, r, m, z)]. \tag{5}$$

Expanding the expectation, we obtain:

$$\mathcal{C}_{\ell_{\mathrm{def}}}(g, r, x) = \mathbb{E}_{y,t|x}\left[ \sum_{j=0}^J c_j(g(x), m_j(x), z) 1_{r(x)=j} \right]. \tag{6}$$

Using the linearity of expectation, this simplifies to:

$$\mathcal{C}_{\ell_{\mathrm{def}}}(g, r, x) = \sum_{j=0}^J \mathbb{E}_{y,t|x}\left[ c_j(g(x), m_j(x), z) \right] 1_{r(x)=j}. \tag{7}$$

Since we seek the rejector that minimizes the expected loss, the Bayes-conditional risk is given by:

$$\mathcal{C}_{\ell_{\mathrm{def}}}^B(\mathcal{G}, \mathcal{R}, x) = \inf_{g \in \mathcal{G}, r \in \mathcal{R}} \mathbb{E}_{y,t|x}[\ell_{\mathrm{def}}(g, r, m, z)]. \tag{8}$$

Rewriting this expression, we obtain:

$$\mathcal{C}_{\ell_{\mathrm{def}}}^B(\mathcal{G}, \mathcal{R}, x) = \inf_{r \in \mathcal{R}} \mathbb{E}_{y,t|x}\left[ \inf_{g \in \mathcal{G}} c_0(g(x), z) 1_{r(x)=0} + \sum_{j=1}^J c_j(m_j(x), z) 1_{r(x)=j} \right]. \tag{9}$$

This leads to the following minimization problem:

$$\mathcal{C}_{\ell_{\mathrm{def}}}^B(\mathcal{G}, \mathcal{R}, x) = \min \left\{ \inf_{g \in \mathcal{G}} \mathbb{E}_{y,t|x}\left[ c_0(g(x), z) \right], \min_{j \in [J]} \mathbb{E}_{y,t|x}\left[ c_j(m_j(x), z) \right] \right\}. \tag{10}$$

To simplify notation, we define:

$$\bar{c}_j^* = \begin{cases} \inf_{g \in \mathcal{G}} \mathbb{E}_{y,t|x}[c_0(g(x), z)], & \text{if } j = 0, \\ \mathbb{E}_{y,t|x}[c_j(m_j(x), z)], & \text{otherwise.} \end{cases} \tag{11}$$

Thus, the Bayes-conditional risk simplifies to:

$$\mathcal{C}_{\ell_{\text{def}}}^B(\mathcal{G}, \mathcal{R}, x) = \min_{j \in \mathcal{A}} \bar{c}_j^*. \tag{12}$$

Since the rejector selects the decision with the lowest expected cost, the optimal rejector is given by:

$$r^B(x) = \begin{cases} 0, & \text{if } \inf_{g \in \mathcal{G}} \mathbb{E}_{y,t|x}[c_0(g(x), z)] \leq \min_{j \in [J]} \mathbb{E}_{y,t|x}[c_j(m_j(x), z)], \\ j, & \text{otherwise.} \end{cases} \tag{13}$$

This completes the proof. $\qquad\square$

## C. Proof Theorem 4.4

Before proving the desired Theorem 4.4, we will use the following Lemma C.1 (Awasthi et al., 2022; Mao et al., 2024h):

**Lemma C.1** ($\mathcal{R}$-consistency bound). *Assume that the following $\mathcal{R}$-consistency bounds holds for $r \in \mathcal{R}$, and any distribution*

$$\mathcal{E}_{\ell_{01}}(r) - \mathcal{E}_{\ell_{01}}^*(\mathcal{R}) + \mathcal{U}_{\ell_{01}}(\mathcal{R}) \leq \Gamma^\nu(\mathcal{E}_{\Phi_{01}^\nu}(r) - \mathcal{E}_{\Phi_{01}^\nu}^*(\mathcal{R}) + \mathcal{U}_{\Phi_{01}^\nu}(\mathcal{R}))$$

*then for $p \in (p_0 \dots p_J) \in \Delta^{|\mathcal{A}|}$ and $x \in \mathcal{X}$, we get*

$$\sum_{j=0}^J p_j \mathbb{1}_{r(x) \neq j} - \inf_{r \in \mathcal{R}} \sum_{j=0}^J p_j \mathbb{1}_{r(x) \neq j} \leq \Gamma^\nu \Big( \sum_{j=0}^J p_j \Phi_{01}^\nu(r, x, j) - \inf_{r \in \mathcal{R}} \sum_{j=0}^J p_j \Phi_{01}^\nu(r, x, j) \Big)$$

**Theorem 4.4** (($\mathcal{G}, \mathcal{R}$)-consistency bounds). *Let $g \in \mathcal{G}$ be a multi-task model. Suppose there exists a non-decreasing function $\Gamma^\nu : \mathbb{R}_+ \to \mathbb{R}_+$, parameterized by $\nu \geq 0$, such that the $\mathcal{R}$-consistency bound holds for any distribution $\mathcal{D}$:*

$$\mathcal{E}_{\Phi_{01}^\nu}(r) - \mathcal{E}_{\Phi_{01}^\nu}^*(\mathcal{R}) + \mathcal{U}_{\Phi_{01}^\nu}(\mathcal{R}) \geq$$
$$\Gamma^\nu \left( \mathcal{E}_{\ell_{01}}(r) - \mathcal{E}_{\ell_{01}}^B(\mathcal{R}) + \mathcal{U}_{\ell_{01}}(\mathcal{R}) \right),$$

*then for any $(g, r) \in \mathcal{G} \times \mathcal{R}$, any distribution $\mathcal{D}$, and any $x \in \mathcal{X}$,*

$$\mathcal{E}_{\ell_{\text{def}}}(g, r) - \mathcal{E}_{\ell_{\text{def}}}^B(\mathcal{G}, \mathcal{R}) + \mathcal{U}_{\ell_{\text{def}}}(\mathcal{G}, \mathcal{R}) \leq$$
$$\overline{\Gamma}^\nu \left( \mathcal{E}_{\Phi_{\text{def}}^\nu}(r) - \mathcal{E}_{\Phi_{\text{def}}^\nu}^*(\mathcal{R}) + \mathcal{U}_{\Phi_{\text{def}}^\nu}(\mathcal{R}) \right)$$
$$+ \mathcal{E}_{c_0}(g) - \mathcal{E}_{c_0}^B(\mathcal{G}) + \mathcal{U}_{c_0}(\mathcal{G}),$$

*where the expected aggregated cost vector is given by $\overline{\tau} = \left( \mathbb{E}_{y,t|x}[\tau_0], \dots, \mathbb{E}_{y,t|x}[\tau_J] \right)$, and $\overline{\Gamma}^\nu(u) = \|\overline{\tau}\|_1 \Gamma^\nu \left( \frac{u}{\|\overline{\tau}\|_1} \right)$ with $\Gamma^\nu(u) = \mathcal{T}^{-1,\nu}(u)$. In the case of the log-softmax surrogate ($\nu = 1$), the transformation is given by $\mathcal{T}^{\nu=1}(u) = \frac{1+u}{2} \log(1+u) + \frac{1-u}{2} \log(1-u)$.*

*Proof.* Let denote a cost for $j \in \mathcal{A} = \{0, \dots, J\}$:

$$\bar{c}_j^* = \begin{cases} \inf_{g \in \mathcal{G}} \mathbb{E}_{y,t|x}[c_0(g(x), z)] & \text{if } j = 0 \\ \mathbb{E}_{y,t|x}[c_j(m(x), z)] & \text{otherwise} \end{cases}$$

Using the change of variables and the Bayes-conditional risk introduced in the proof of Lemma 4.2 in Appendix B, we have:

$$\mathcal{C}_{\ell_{\text{def}}}^B(\mathcal{G}, \mathcal{R}, x) = \min_{j \in \mathcal{A}} \bar{c}_j^*$$

$$\mathcal{C}_{\ell_{\text{def}}}(g, r, x) = \sum_{j=0}^J \mathbb{E}_{y,t|x} \Big[ c_j(g(x), m_j(x), z) \Big] \mathbb{1}_{r(x)=j} \tag{14}$$

We follow suit for our surrogate $\Phi_{\text{def}}$ and derive its conditional risk and optimal conditional risk.

$$\mathcal{C}_{\Phi_{\text{def}}} = \mathbb{E}_{y,t|x}\Big[ \sum_{j=1}^{J} c_j(m(x),z)\Phi_{01}^{\nu}(r,x,0) + \sum_{j=1}^{J}\Big( c_0(g(x),z) + \sum_{i=1}^{J} c_i(m_i(x),z)1_{j\neq i}\Big)\Phi_{01}^{\nu}(r,x,j)\Big]$$

$$\mathcal{C}_{\Phi_{\text{def}}}^{*} = \inf_{r\in\mathcal{R}} \mathbb{E}_{y,t|x}\Big[ \sum_{j=1}^{J} c_j(g(x),m(x),z)\Phi_{01}^{\nu}(r,x,0) + \sum_{j=1}^{J}[c_0(g(x),z) + \sum_{i=1}^{J} c_i(m_i(x),z)1_{j\neq i}]\Phi_{01}^{\nu}(r,x,j)\Big]$$

Let us define the function $v(m(x),z) = \min_{j\in[J]} \overline{c}_j(m_j(x),z)$, where $m_j(x)$ denotes the model's output and $\overline{c}_j$ represents the corresponding cost function. Using this definition, the calibration gap is formulated as $\Delta\mathcal{C}_{\ell_{\text{def}}} := \mathcal{C}_{\ell_{\text{def}}} - \mathcal{C}_{\ell_{\text{def}}}^{B}$, where $\mathcal{C}_{\ell_{\text{def}}}$ represents the original calibration term and $\mathcal{C}_{\ell_{\text{def}}}^{B}$ denotes the baseline calibration term. By construction, the calibration gap satisfies $\Delta\mathcal{C}_{\ell_{\text{def}}} \geq 0$, leveraging the risks derived in the preceding analysis.

$$\Delta\mathcal{C}_{\ell_{\text{def}}} = \mathbb{E}_{y,t|x}\Big[ \rho(g(x),z)1_{r(x)=0} + \sum_{j=1}^{J}\Big( \rho(m(x),z) + \beta_j\Big)1_{r(x)=j}\Big]$$
$$- v(m(x),z) + \Big( v(m(x),z) - \min_{j\in\mathcal{A}} \overline{c}_j^{*}(g(x),m(x),z)\Big)$$

Let us consider $\Delta\mathcal{C}_{\ell_{\text{def}}} = A_1 + A_2$, such that:

$$A_1 = \mathbb{E}_{y,t|x}\Big[ 1_{r(x)=0}\rho(g(x),z) + \sum_{j=1}^{J} 1_{r(x)=j}\Big( \rho(m_j(x),z) + \beta_j\Big)\Big] - v(m(x),z)$$
$$A_2 = \Big( v(m(x),z) - \min_{j\in\mathcal{A}} \overline{c}_j(g(x),m(x),z)\Big) \tag{15}$$

By considering the properties of $\min$, we also get the following inequality:

$$v(m(x),z) - \min_{j\in\mathcal{A}} \overline{c}_j^{*}(g(x),m(x),z) \leq \mathbb{E}_{y,t|x}[c_0(g(x),z)] - \inf_{g\in\mathcal{G}} \mathbb{E}_{y,t|x}[c_0(g(x),z)] \tag{16}$$

implying,

$$\Delta\mathcal{C}_{\ell_{\text{def}}} \leq A_1 + \overline{c}_0(g(x),z) - \overline{c}_0^{*}(g(x),z) \tag{17}$$

We now select a distribution for our rejector. We first define $\forall j \in \mathcal{A}$,

$$p_0 = \frac{\sum_{j=1}^{J} \overline{c}_j(m_j(x),z)}{J\sum_{j=0}^{J} \overline{c}_j(g(x),m_j(x),z)}$$

and

$$p_{j\in[J]} = \frac{\overline{c}_0(g(x),z) + \sum_{j\neq j'}^{J} \overline{c}_j'(m_j(x),z)}{J\sum_{j=0}^{J} \overline{c}_j(g(x),m_j(x),z)}$$

which can also be written as:

$$p_j = \frac{\overline{\tau}_j}{\|\overline{\tau}\|_1} \tag{18}$$

Injecting the new distribution, we obtain the following:

$$\Delta\mathcal{C}_{\Phi_{\text{def}}} = \|\overline{\tau}\|_1\Big( \sum_{j=0}^{J} p_j\Phi_{01}^{\nu}(r,x,j) - \inf_{r\in\mathcal{R}} \sum_{j=0}^{J} p_j\Phi_{01}^{\nu}(r,x,j)\Big) \tag{19}$$

Now consider the first and last term of $\Delta\mathcal{C}_{\ell_{\text{def}}}$. Following the intermediate step for Lemma 4.3, we have:

$$A_1 = \mathbb{E}_{y,t|x}[c_0(g(x),z)]1_{r(x)=0} + \sum_{j=1}^{J}\mathbb{E}_{y,t|x}[c_j(m_j(x),z)]1_{r(x)=j} - v(m(x),z)$$

$$= \mathbb{E}_{y,t|x}[c_0(g(x),z)]1_{r(x)=0} + \sum_{j=1}^{J}\mathbb{E}_{y,t|x}[c_j(m_j(x),z)]1_{r(x)=j}$$

$$- \inf_{r\in\mathcal{R}}\left[\mathbb{E}_{y,t|x}[c_0(g(x),z)]1_{r(x)=0} + \sum_{j=1}^{J}\mathbb{E}_{y,t|x}[c_j(m_j(x),z)]1_{r(x)=j}\right]$$

$$= \sum_{j=1}^{J}\overline{c}_j(z,m_j)1_{r(x)\neq 0} + \sum_{j=1}^{J}\left(\overline{c}_0(g(x),z) + \sum_{j\neq j'}^{J}\overline{c}_{j'}(m_{j'}(x),z)\right)1_{r(x)\neq j}$$

$$- \inf_{r\in\mathcal{R}}\left[\sum_{j=1}^{J}\overline{c}_j(m_{j'}(x),z)1_{r(x)\neq 0} + \sum_{j=1}^{J}\left(\overline{c}_0(g(x),z) + \sum_{j\neq j'}^{J}\overline{c}_{j'}(m_{j'}(x),z)\right)1_{r(x)\neq j}\right]$$

Then, applying a change of variables to introduce $\|\overline{\tau}\|_1$, we get:

$$\|\overline{\tau}\|_1 p_0 1_{r(x)\neq 0} + \|\overline{\tau}\|_1 \sum_{j=1}^{J} p_j 1_{r(x)\neq j} - \inf_{r\in\mathcal{R}}[\|\overline{\tau}\|_1 p_0 1_{r(x)\neq 0} + \|\overline{\tau}\|_1 \sum_{j=1}^{J} p_j 1_{r(x)\neq j}]$$

$$= \|\overline{\tau}\|_1 \sum_{j=0}^{J} p_j 1_{r(x)\neq j} - \inf_{r\in\mathcal{R}}\|\overline{\tau}\|_1 \sum_{j=0}^{J} p_j 1_{r(x)\neq j}$$

We now apply Lemma C.1 to introduce $\Gamma$,

$$\sum_{j=0}^{J} p_j 1_{r(x)\neq j} - \inf_{r\in\mathcal{R}}\sum_{j=0}^{J} p_j 1_{r(x)\neq j} \leq \Gamma\left(\sum_{j=0}^{J} p_j \Phi_{01}^{\nu}(r,x,j) - \inf_{r\in\mathcal{R}}\sum_{j=0}^{J} p_j \Phi_{01}^{\nu}(r,x,j)\right)$$

$$\frac{1}{\|\overline{\tau}\|_1}\left[\sum_{j=0}^{J}\overline{\tau}_j 1_{r(x)\neq j} - \inf_{r\in\mathcal{R}}\sum_{j=0}^{J}\overline{\tau}_j 1_{r(x)\neq j}\right] \leq \Gamma\left(\frac{1}{\|\overline{\tau}\|_1}\left[\sum_{j=0}^{J}\overline{\tau}_j \Phi_{01}^{\nu}(r,x,j) - \inf_{r\in\mathcal{R}}\sum_{j=0}^{J}\overline{\tau}_j \Phi_{01}^{\nu}(r,x,j)\right]\right) \qquad (20)$$

$$\Delta\mathcal{C}_{\ell_{\text{def}}} \leq \|\overline{\tau}\|_1 \Gamma\left(\frac{\Delta\mathcal{C}_{\Phi_{\text{def}}}}{\|\overline{\tau}\|_1}\right)$$

We reintroduce the coefficient $A_2$ such that:

$$\Delta\mathcal{C}_{\ell_{\text{def}}} \leq \|\overline{\tau}\|_1 \Gamma\left(\frac{\Delta\mathcal{C}_{\Phi_{\text{def}}}}{\|\overline{\tau}\|_1}\right) + A_2$$

$$\Delta\mathcal{C}_{\ell_{\text{def}}} \leq \|\overline{\tau}\|_1 \Gamma\left(\frac{\Delta\mathcal{C}_{\Phi_{\text{def}}}}{\|\overline{\tau}\|_1}\right) + \mathbb{E}_{y,t|x}[c_0(g(x),z)] - \inf_{g\in\mathcal{G}}\mathbb{E}_{y,t|x}[c_0(g(x),z)] \quad \text{(upper bounding with Eq 16)}$$

Mao et al. (2023b) introduced a tight bound for the comp-sum surrogates family. It follows for $\nu \geq 0$ the inverse transformation $\Gamma^{\nu}(u) = \mathcal{T}^{-1,\nu}(u)$:

$$\mathcal{T}^{\nu}(v) = \begin{cases} \frac{2^{1-\nu}}{1-\nu}\left[1 - \left(\frac{(1+v)^{\frac{2-\nu}{2}} + (1-v)^{\frac{2-\nu}{2}}}{2}\right)^{2-\nu}\right] & \nu \in [0,1) \\[3mm] \frac{1+v}{2}\log[1+v] + \frac{1-v}{2}\log[1-v] & \nu = 1 \\[3mm] \frac{1}{(\nu-1)n^{\nu-1}}\left[\left(\frac{(1+v)^{\frac{2-\nu}{2}} + (1-v)^{\frac{2-\nu}{2}}}{2}\right)^{2-\nu} - 1\right] & \nu \in (1,2) \\[3mm] \frac{1}{(\nu-1)n^{\nu-1}}v & \nu \in [2,+\infty). \end{cases}$$

We note $\overline{\Gamma}^\nu(u) = \|\overline{\tau}\|_1 \Gamma^\nu(\frac{u}{\|\overline{\tau}\|_1})$. By applying Jensen's Inequality and taking expectation on both sides, we get

$$\mathcal{E}_{\ell_{\text{def}}}(g,r) - \mathcal{E}^B_{\ell_{\text{def}}}(\mathcal{G},\mathcal{R}) + \mathcal{U}_{\ell_{\text{def}}}(\mathcal{G},\mathcal{R})$$
$$\leq \overline{\Gamma}^\nu(\mathcal{E}_{\Phi_{\text{def}}}(r) - \mathcal{E}^*_{\Phi_{\text{def}}}(\mathcal{R}) + \mathcal{U}_{\Phi_{\text{def}}}(\mathcal{R})) + \mathcal{E}_{c_0}(g) - \mathcal{E}^B_{c_0}(\mathcal{G}) + \mathcal{U}_{c_0}(\mathcal{G})$$

$\square$

## D. Proof Theorem 4.6

**Theorem 4.6** (Characterization of Minimizability Gaps)**.** *Assume $\mathcal{R}$ is symmetric and complete. Then, for the cross-entropy multiclass surrogates $\Phi^\nu_{01}$ and any distribution $\mathcal{D}$, the following holds for $\nu \geq 0$:*

$$\mathcal{C}^{\nu,*}_{\Phi^\nu_{\text{def}}}(\mathcal{R},x) = \begin{cases} \|\overline{\tau}\|_1 H\left(\frac{\overline{\tau}}{\|\overline{\tau}\|_1}\right) & \text{for } \nu = 1 \\ \|\overline{\tau}\|_1 - \|\overline{\tau}\|_\infty & \nu = 2 \\ \frac{1}{\nu-1}\left[\|\overline{\tau}\|_1 - \|\overline{\tau}\|_{\frac{1}{2-\nu}}\right] & \nu \in (1,2) \\ \frac{1}{1-\nu}\left[\left(\sum_{k=0}^J \overline{\tau}_k^{\frac{1}{2-\nu}}\right)^{2-\nu} - \|\overline{\tau}\|_1\right] & \nu > 2, \end{cases}$$

*where $\overline{\tau} = \{\mathbb{E}_{y,t|x}[\overline{\tau}_0], \ldots, \mathbb{E}_{y,t|x}[\overline{\tau}_J]\}$, the aggregated costs are $\tau_j = \sum_{k=0}^J c_k 1_{k\neq j}$, and $H$ denotes the Shannon entropy. The minimizability gap is defined as $\mathcal{U}_{\Phi^\nu_{\text{def}}}(\mathcal{R}) = \mathcal{E}^*_{\Phi^\nu_{\text{def}}}(\mathcal{R}) - \mathbb{E}_x\left[\mathcal{C}^{\nu,*}_{\Phi^\nu_{\text{def}}}(\mathcal{R},x)\right]$.*

*Proof.* We define the softmax distribution as $s_j = \frac{e^{r(x,j)}}{\sum_{j'\in\mathcal{A}} e^{r(x,j')}}$, where $s_j \in [0,1]$. Let $\overline{\tau}_j = \tau_j(g(x), m(x), z)$ with $\tau_j \in \mathbb{R}^+$, and denote the expected value as $\overline{\tau} = \mathbb{E}_{y,t|x}[\tau]$. We now derive the conditional risk for a given $\nu \geq 0$:

$$\mathcal{C}^\nu_{\Phi_{\text{def}}}(r,x) = \sum_{j=0}^J \mathbb{E}_{y,t|x}[\tau_j]\Phi^\nu_{01}(r,x,j)$$
$$= \begin{cases} \frac{1}{1-\nu}\sum_{j=0}^J \overline{\tau}_j\left[\left(\sum_{j'\in\mathcal{A}} e^{r(x,j')-r(x,j)}\right)^{1-\nu} - 1\right] & \nu \neq 1 \\ \sum_{j=0}^J \overline{\tau}_j \log\left(\sum_{j'\in\mathcal{A}} e^{r(x,j')-r(x,j)}\right) & \nu = 1 \end{cases} \quad (21)$$
$$= \begin{cases} \frac{1}{1-\nu}\sum_{j=0}^J \overline{\tau}_j\left[s_j^{\nu-1} - 1\right] & \nu \neq 1 \\ -\sum_{j=0}^J \overline{\tau}_j \log(s_j) & \nu = 1 \end{cases}$$

**For $\nu = 1$:** we can write the following conditional risk:

$$\mathcal{C}^{\nu=1}_{\Phi_{\text{def}}}(r,x) = -\sum_{j=0}^J \overline{\tau}_j\left[r(x,j) - \log\sum_{j'\in\mathcal{A}} e^{r(x,j')}\right] \quad (22)$$

Then,

$$\frac{\partial\mathcal{C}^{\nu=1}_{\Phi_{\text{def}}}}{\partial r(x,i)}(r,x) = -\overline{\tau}_i + \left(\sum_{j=0}^J \overline{\tau}_j\right)s_i^* \quad (23)$$

At the optimum, we have:

$$s^*(x,i) = \frac{\overline{\tau}_i}{\sum_{j=0}^J \overline{\tau}_j} \quad (24)$$

Then, it follows:

$$\mathcal{C}^{*,\nu=1}_{\Phi_{\text{def}}}(\mathcal{R},x) = -\sum_{j=0}^J \overline{\tau}_j \log\left(\frac{\overline{\tau}_j}{\sum_{j'=0}^J \overline{\tau}_{j'}}\right) \quad (25)$$

As the softmax parametrization is a distribution $s^* \in \Delta^{|\mathcal{A}|}$, we can write this conditional in terms of entropy with $\overline{\tau} = \{\overline{\tau}_j\}_{j \in \mathcal{A}}$:

$$
\begin{aligned}
\mathcal{C}^{*,\nu=1}_{\Phi_{\text{def}}}(\mathcal{R}, x) &= -\Big(\sum_{k=0}^{J} \overline{\tau}_k\Big) \sum_{j=0} s_j^* \log(s_j^*) \\
&= \Big(\sum_{k=0}^{J} \overline{\tau}_k\Big) H\Big(\frac{\overline{\tau}}{\sum_{j'=0} \overline{\tau}_{j'}}\Big) \\
&= \|\overline{\tau}\|_1 H\Big(\frac{\overline{\tau}}{\|\overline{\tau}\|_1}\Big) \quad (\text{as } \tau_j \in \mathbb{R}^+)
\end{aligned}
\tag{26}
$$

**For $\nu \neq 1, 2$:** The softmax parametrization can be written as a constraint $\sum_{j=0}^{J} s_j = 1$ and $s_j \geq 0$. Consider the objective

$$
\Phi(\mathbf{s}) = \frac{1}{1-\nu} \sum_{j=0}^{J} \overline{\tau}_j \left[ s_j^{\nu-1} - 1 \right].
\tag{27}
$$

We aim to find $\mathbf{s}^* = (s_0^*, \dots, s_J^*)$ that minimizes (27) subject to $\sum_{j=0}^{J} s_j = 1$. Introduce a Lagrange multiplier $\lambda$ for the normalization $\sum_{j=0}^{J} s_j = 1$. The Lagrangian is:

$$
\mathcal{L}(\mathbf{s}, \lambda) = \frac{1}{1-\nu} \sum_{j=0}^{J} \overline{\tau}_j \left[ s_j^{\nu-1} - 1 \right] + \lambda\Big(1 - \sum_{j=0}^{J} s_j\Big).
\tag{28}
$$

We take partial derivatives with respect to $s_i$:

$$
\frac{\partial \mathcal{L}}{\partial s_i} = \frac{1}{1-\nu} \overline{\tau}_i (\nu - 1) s_i^{\nu-2} - \lambda = 0.
\tag{29}
$$

Since $\frac{\nu-1}{1-\nu} = -1$, we get

$$
\overline{\tau}_i s_i^{\nu-2} = -\lambda > 0 \implies s_i^{\nu-2} = \frac{\alpha}{\overline{\tau}_i} \quad \text{for some } \alpha > 0.
\tag{30}
$$

Hence

$$
s_i = \Big(\frac{\alpha}{\overline{\tau}_i}\Big)^{\frac{1}{\nu-2}}.
\tag{31}
$$

Summing $s_i$ over $\{i = 0, \dots, J\}$ and setting the total to 1 yields:

$$
\sum_{i=0}^{J} \Big(\frac{\alpha}{\overline{\tau}_i}\Big)^{\frac{1}{\nu-2}} = 1.
\tag{32}
$$

Let

$$
\alpha^{\frac{1}{\nu-2}} = \frac{1}{\sum_{k=0}^{J}\big(\frac{1}{\overline{\tau}_k}\big)^{\frac{1}{\nu-2}}} \implies \alpha = \Big[\sum_{k=0}^{J}\big(\frac{1}{\overline{\tau}_k}\big)^{\frac{1}{\nu-2}}\Big]^{\nu-2}.
\tag{33}
$$

Therefore, for each $i$,

$$
s_i^* = \Big(\frac{\alpha}{\overline{\tau}_i}\Big)^{\frac{1}{\nu-2}} = \frac{\overline{\tau}_i^{\frac{1}{2-\nu}}}{\sum_{k=0}^{J} \overline{\tau}_k^{\frac{1}{2-\nu}}}.
\tag{34}
$$

This $\{s_i^*\}$ is a valid probability distribution. Let

$$
A = \sum_{k=0}^{J} \tau_k^{\frac{1}{2-\nu}}.
\tag{35}
$$

Then the optimum distribution is

$$s_i^* = \frac{\overline{\tau}_i^{\frac{1}{2-\nu}}}{A}. \tag{36}$$

Recall

$$\Phi(\mathbf{s}) = \frac{1}{1-\nu} \sum_{j=0}^{J} \overline{\tau}_j \left[ s_j^{\nu-1} - 1 \right]. \tag{37}$$

At $s_j^*$, we have

$$(s_j^*)^{\nu-1} = \left( \frac{\overline{\tau}_j^{\frac{1}{2-\nu}}}{A} \right)^{\nu-1} = \frac{\overline{\tau}_j^{\frac{\nu-1}{2-\nu}}}{A^{\nu-1}}. \tag{38}$$

Hence

$$\sum_{j=0}^{J} \overline{\tau}_j \left( s_j^* \right)^{\nu-1} = \frac{1}{A^{\nu-1}} \sum_{j=0}^{J} \overline{\tau}_j^{1+\frac{\nu-1}{2-\nu}} = \frac{1}{A^{\nu-1}} \sum_{j=0}^{J} \overline{\tau}_j^{\frac{1}{2-\nu}} = \frac{A}{A^{\nu-1}} = A^{2-\nu}. \tag{39}$$

Substituting back,

$$\mathcal{C}_{\Phi_{\mathrm{def}}}^{*,\nu\neq1,2}(\mathcal{R},x) = \frac{1}{1-\nu} \left[ \left( \sum_{k=0}^{J} \overline{\tau}_k^{\frac{1}{2-\nu}} \right)^{2-\nu} - \sum_{j=0}^{J} \overline{\tau}_j \right] \tag{40}$$

We can express this conditional risk with a valid $L^{\left( \frac{1}{2-\nu} \right)}$ norm as long as $\nu \in (1,2)$.

$$\mathcal{C}_{\Phi_{\mathrm{def}}}^{*,\nu\neq1,2}(\mathcal{R},x) = \frac{1}{\nu-1} \left[ \|\overline{\boldsymbol{\tau}}\|_1 - \|\overline{\boldsymbol{\tau}}\|_{\frac{1}{2-\nu}} \right] \tag{41}$$

**For $\nu = 2$:** Since $\sum_{j=0}^{J} \overline{\tau}_j = S$, we have

$$\mathcal{C}_{\Phi_{\mathrm{def}}}^{\nu=2}(r,x) = \sum_{j=0}^{J} \overline{\tau}_j \left[ 1 - s_j(r) \right] = \sum_{j=0}^{J} \overline{\tau}_j - \sum_{j=0}^{J} \overline{\tau}_j \, s_j(r). \tag{42}$$

Hence

$$\inf_{r \in \mathcal{R}} \mathcal{C}_{\Phi_{\mathrm{def}}}^{\nu=2}(r,x) = S - \sup_{r \in \mathcal{R}} \sum_{j=0}^{J} \overline{\tau}_j \, s_j(r). \tag{43}$$

Therefore, minimizing $\mathcal{C}_{\Phi_{\mathrm{def}}}^{\nu=2}(r,x)$ is equivalent to maximizing

$$F(r) = \sum_{j=0}^{J} \overline{\tau}_j \, s_j(r). \tag{44}$$

Its partial derivative w.r.t. $r_i$ is the standard softmax derivative:

$$\frac{\partial s_j}{\partial r_i} = s_j \left( \delta_{ij} - s_i \right) = \begin{cases} s_i \left( 1 - s_i \right), & \text{if } i = j, \\ -s_j \, s_i, & \text{otherwise.} \end{cases} \tag{45}$$

Hence, for each $i$,

$$\frac{\partial F}{\partial r_i} = \sum_{j=0}^{J} \overline{\tau}_j \frac{\partial s_j}{\partial r_i} = \overline{\tau}_i \, s_i \left( 1 - s_i \right) + \sum_{\substack{j=0 \\ j\neq i}}^{J} \overline{\tau}_j \left( -s_j \, s_i \right). \tag{46}$$

Factor out $s_i$:

$$\frac{\partial F}{\partial r_i} = s_i \left[ \overline{\tau}_i \left( 1 - s_i \right) - \sum_{j\neq i} \overline{\tau}_j \, s_j \right] = s_i \left[ \overline{\tau}_i - \left( \sum_{j=0}^{J} \overline{\tau}_j \, s_j \right) \right], \tag{47}$$

because $\sum_{j\neq i} \overline{\tau}_j s_j = \sum_{j=0}^{J} \overline{\tau}_j s_j - \overline{\tau}_i s_i$. Define $F(r) = \sum_{j=0}^{J} \overline{\tau}_j s_j(r)$. Then:

$$\frac{\partial F}{\partial r_i} = s_i \left[ \overline{\tau}_i - F(r) \right]. \tag{48}$$

Setting $\frac{\partial F}{\partial r_i} = 0$ for each $i$ implies

$$s_i \left[ \overline{\tau}_i - F(r) \right] = 0, \quad \forall i. \tag{49}$$

Thus, for each index $i$:

$$s_i = 0 \quad \text{or} \quad \overline{\tau}_i = F(r). \tag{50}$$

To maximize $F(r)$, notice that:

- If $\overline{\tau}_{i^*}$ is strictly the largest among all $\overline{\tau}_i$, then the maximum is approached by making $s_{i^*} \approx 1$, so $F(r) \approx \overline{\tau}_{i^*}$. In the softmax parameterization, this occurs in the limit $r_{i^*} \to +\infty$ and $r_k \to -\infty$ for $k \neq i^*$.

- If there is a tie for the largest $\overline{\tau}_i$, we can put mass on those coordinates that share the maximum value. In any case, the supremum is $\max_i \overline{\tau}_i$.

Hence

$$\sup_{r \in \mathcal{R}} F(r) = \max_{0 \leq i \leq J} \overline{\tau}_i. \tag{51}$$

Because $\mathcal{C}_{\Phi_{\text{def}}}^{\nu=2}(r, x) = S - F(r)$,

$$\inf_{r \in \mathcal{R}} \mathcal{C}_{\Phi_{\text{def}}}^{\nu=2}(r, x) = S - \sup_{r \in \mathcal{R}} F(r) = \sum_{j=0}^{J} \overline{\tau}_j - \max_{i \in \mathcal{A}} \overline{\tau}_i = \|\overline{\tau}\|_1 - \|\overline{\tau}\|_\infty \tag{52}$$

Hence the global minimum of $\mathcal{C}_{\Phi_{\text{def}}}^{\nu=2}$ is $\|\overline{\tau}\|_1 - \|\overline{\tau}\|_\infty$. In the "softmax" parameterization, this is only approached in the limit as one coordinate $r_{i^*}$ goes to $+\infty$ and all others go to $-\infty$. No finite $r$ yields an exactly one-hot $s_i(r) = 1$, but the limit is enough to achieve the infimum arbitrarily closely.

It follows for $\overline{\tau} = \{\overline{\tau}_j\}_{j \in \mathcal{A}}$ and $\nu \geq 0$:

$$\inf_{r \in \mathcal{R}} \mathcal{C}_{\Phi_{\text{def}}}^{\nu}(r, x) = \begin{cases} \|\overline{\tau}\|_1 H\left( \frac{\overline{\tau}}{\|\overline{\tau}\|_1} \right) & \nu = 1 \\ \|\overline{\tau}\|_1 - \|\overline{\tau}\|_\infty & \nu = 2 \\ \frac{1}{\nu-1} \left[ \|\overline{\tau}\|_1 - \|\overline{\tau}\|_{\frac{1}{2-\nu}} \right] & \nu \in (1, 2) \\ \frac{1}{1-\nu} \left[ \left( \sum_{k=0}^{J} \overline{\tau}_k^{\frac{1}{2-\nu}} \right)^{2-\nu} - \|\overline{\tau}\|_1 \right] & \text{otherwise} \end{cases} \tag{53}$$

Building on this, we can infer the minimizability gap:

$$\mathcal{U}_{\Phi_{\text{def}}}(\mathcal{R}) = \mathcal{E}_{\Phi_{\text{def}}}^*(\mathcal{R}) - \mathbb{E}_x \left[ \inf_{r \in \mathcal{R}} \mathcal{C}_{\Phi_{\text{def}}}^{\nu}(r, x) \right] \tag{54}$$

$\square$

# E. Proof Lemma 4.12

**Lemma 4.12.** *Let $\mathcal{L}_1$ be a family of functions mapping $\mathcal{X}$ to $[0, 1]$, and let $\mathcal{L}_2$ be a family of functions mapping $\mathcal{X}$ to $\{0, 1\}$. Define $\mathcal{L} = \{l_1 l_2 : l_1 \in \mathcal{L}_1, l_2 \in \mathcal{L}_2\}$. Then, the empirical Rademacher complexity of $\mathcal{L}$ for any sample $S$ of size $K$ is bounded by:*

$$\widehat{\mathfrak{R}}_S(\mathcal{L}) \leq \widehat{\mathfrak{R}}_S(\mathcal{L}_1) + \widehat{\mathfrak{R}}_S(\mathcal{L}_2).$$

*Proof.* We define the function $\psi$ as follows:

$$\psi : \begin{array}{ccc} \mathcal{L}_1 + \mathcal{L}_2 & \longrightarrow & \mathcal{L}_1\mathcal{L}_2 \\ l_1 + l_2 & \longmapsto & (l_1 + l_2 - 1)_+ \end{array} \tag{55}$$

Here, $l_1 \in \mathcal{L}_1$ and $l_2 \in \mathcal{L}_2$. The function $\psi$ is 1-Lipschitz as we have $t \mapsto (t-1)_+$ for $t = l_1 + l_2$. Furthermore, given that $\psi$ is surjective and 1-Lipschitz, by Talagrand's lemma (Mohri et al., 2012), we have:

$$\hat{\mathfrak{R}}_S(\psi(\mathcal{L}_1 + \mathcal{L}_2)) \le \hat{\mathfrak{R}}_S(\mathcal{L}_1 + \mathcal{L}_2) \le \hat{\mathfrak{R}}_S(\mathcal{L}_1) + \hat{\mathfrak{R}}_S(\mathcal{L}_2) \tag{56}$$

This inequality shows that the Rademacher complexity of the sum of the losses is bounded by the sum of their individual complexities. $\square$

## F. Proof Theorem 4.13

**Theorem 4.13** (Learning bounds of the deferral loss). *For any expert $M_j$, any distribution $\mathcal{D}$ over $\mathcal{Z}$, we have with probability $1 - \delta$ for $\delta \in [0, 1/2]$, that the following bound holds at the optimum:*

$$\mathcal{E}_{\ell_{def}}(h, f, r) \le \widehat{\mathcal{E}}_{\ell_{def}}(h, f, r) + 2\mathfrak{R}_K(\mathcal{L}_{def}) + \sqrt{\frac{\log 1/\delta}{2K}},$$

*with*

$$\mathfrak{R}_K(\mathcal{L}_{def}) \le \frac{1}{2}\mathfrak{R}_K(\mathcal{H}) + \mathfrak{R}_K(\mathcal{F}) + \sum_{j=1}^{J} \Omega(m_j^h, y)$$

$$+ \Big( \sum_{j=1}^{J} \max \ell_{reg}(m_j^f, t) + 2 \Big) \mathfrak{R}_K(\mathcal{R}),$$

*with $\Omega(m_j^h, y) = \frac{1}{2}\mathcal{D}(m_j^h \ne y) \exp\left(-\frac{K}{8}\mathcal{D}(m_j^h \ne y)\right) + \mathcal{R}_{K\mathcal{D}(m_j^h \ne y)/2}(\mathcal{R}).$*

*Proof.* We are interested in finding the generalization of $u = (g, r) \in \mathcal{L}$:

$$\mathfrak{R}_S(\mathcal{L}) = \frac{1}{K}\mathbb{E}_\sigma[\sup_{g \in \mathcal{L}} \sum_{k=1}^{K} \sigma_k \ell_{def}(g, r, x_k, y_k, b_k, m_k)]$$

$$= \frac{1}{K}\mathbb{E}_\sigma[\sup_{g \in \mathcal{L}} \sum_{k=1}^{K} \sigma_k \Big( \sum_{j=0}^{J} c_j 1_{r(x_k)=j} \Big)]$$

$$\le \frac{1}{K}\mathbb{E}_\sigma\Big[ \sup_{g \in \mathcal{L}} \sum_{k=1}^{K} \sigma_k c_0 1_{r(x_k)=0} \Big] + \frac{1}{K}\sum_{j=1}^{J} \mathbb{E}_\sigma\Big[ \sup_{r \in \mathcal{R}} \sum_{k=1}^{K} \sigma_k c_j 1_{r(x_k)=j} \Big] \quad \text{(By the subadditivity of sup)}$$

Let's consider $j = 0$:

$$\frac{1}{K}\mathbb{E}_\sigma\Big[ \sup_{g \in \mathcal{L}} \sum_{k=1}^{K} \sigma_k c_0 1_{r(x_k)=0} \Big] = \frac{1}{K}\mathbb{E}_\sigma\Big[ \sup_{g \in \mathcal{L}} \sum_{k=1}^{K} \sigma_k [1_{h(x_k)\ne y} + \ell_{reg}(f(x_k), b_k)] 1_{r(x_k)=0} \Big]$$

$$\le \frac{1}{K}\mathbb{E}_\sigma\Big[ \sup_{g \in \mathcal{L}} \sum_{k=1}^{K} \sigma_k 1_{h(x_k)\ne y} 1_{r(x_k)=0} \Big] + \frac{1}{K}\mathbb{E}_\sigma\Big[ \sup_{g \in \mathcal{L}} \sum_{k=1}^{K} \sigma_k \ell_{reg}(f(x_k), b_k) 1_{r(x_k)=0} \Big]$$

$$\le \Big[ \frac{1}{2}\mathfrak{R}_K(\mathcal{H}) + \mathfrak{R}_K(\mathcal{R}) \Big] + \Big[ \mathfrak{R}_K(\mathcal{F}) + \mathfrak{R}_K(\mathcal{R}) \Big] \quad \text{(using Lemma 4.12)}$$

$$= \frac{1}{2}\mathfrak{R}_K(\mathcal{H}) + \mathfrak{R}_K(\mathcal{F}) + 2\mathfrak{R}_K(\mathcal{R})$$

$$\tag{57}$$

Let's consider $j > 0$:

$$\frac{1}{K}\sum_{j=1}^{J}\mathbb{E}_\sigma\left[\sup_{r\in\mathcal{R}}\sum_{k=1}^{K}\sigma_k c_j 1_{r(x_k)=j}\right] \leq \frac{1}{K}\sum_{j=1}^{J}\mathbb{E}_\sigma\left[\sup_{r\in\mathcal{R}}\sum_{k=1}^{K}\sigma_k 1_{m^h_{k,j}\neq y}1_{r(x_k)=j}\right]$$
$$+ \frac{1}{K}\sum_{j=1}^{J}\mathbb{E}_\sigma\left[\sup_{r\in\mathcal{R}}\sum_{k=1}^{K}\sigma_k \ell_{\mathrm{reg}}(m^f_{k,j},b_k)1_{r(x_k)=j}\right] \tag{58}$$

Using learning-bounds for single expert in classification (Mozannar & Sontag, 2020), we have:

$$\frac{1}{K}\mathbb{E}_\sigma\left[\sup_{r\in\mathcal{R}}\sum_{k=1}^{K}\sigma_k 1_{m^h_k\neq y}1_{r(x_k)=1}\right] \leq \frac{\mathcal{D}(m^h\neq y)}{2}\exp\left(-\frac{K\mathcal{D}(m^h\neq y)}{8}\right) + \mathcal{R}_{K\mathcal{D}(m^h\neq y)/2}(\mathcal{R}) \tag{59}$$

Applying it to our case:

$$\frac{1}{K}\sum_{j=1}^{J}\mathbb{E}_\sigma\left[\sup_{r\in\mathcal{R}}\sum_{k=1}^{K}\sigma_k 1_{m^h_{k,j}\neq y}1_{r(x_k)=j}\right] \leq \sum_{j=1}^{J}\left(\frac{\mathcal{D}(m^h_j\neq y)}{2}\exp\left(-\frac{K\mathcal{D}(m^h_j\neq y)}{8}\right) + \mathcal{R}_{K\mathcal{D}(m^h_j\neq y)/2}(\mathcal{R})\right) \tag{60}$$

For the last term,

$$\frac{1}{K}\sum_{j=1}^{J}\mathbb{E}_\sigma\left[\sup_{r\in\mathcal{R}}\sum_{k=1}^{K}\sigma_k \ell_{\mathrm{reg}}(m^f_{k,j},b_k)1_{r(x_k)=j}\right] \leq \sum_{j=1}^{J}\left(\max \ell_{reg}(m^f_j,t)\mathfrak{R}_K(\mathcal{R})\right) \tag{61}$$

Then, it leads to:

$$\mathfrak{R}_K(\mathcal{L}_{\mathrm{def}}) \leq \frac{1}{2}\mathfrak{R}_K(\mathcal{H}) + \mathfrak{R}_K(\mathcal{F}) + \sum_{j=1}^{J}\Omega(m^h_j,y) + \left(\sum_{j=1}^{J}\max \ell_{reg}(m^f_j,t) + 2\right)\mathfrak{R}_K(\mathcal{R})$$

with $\Omega(m^h_j,y) = \frac{\mathcal{D}(m^h_j\neq y)}{2}\exp\left(-\frac{K\mathcal{D}(m^h_j\neq y)}{8}\right) + \mathcal{R}_{K\mathcal{D}(m^h_j\neq y)/2}(\mathcal{R})$ $\qquad\qquad\square$

## G. Experiments

### G.1. PascalVOC Experiment

Since an image may contain multiple objects, our deferral rule is applied at the level of the entire image $x \in \mathcal{X}$, ensuring that the approach remains consistent with real-world scenarios.

| | Model | $M_1$ | $M_2$ |
|---|---|---|---|
| mAP | 39.5 | 43.3 | 52.8 |

Table 2. Agent accuracies on the CIFAR-100 validation set. Since the training and validation sets are pre-determined in this dataset, the agents' knowledge remains fixed throughout the evaluation.

| Cost $\beta_2$ | mAP (%) | Model Allocation (%) | Expert 1 Allocation (%) | Expert 2 Allocation (%) |
|---|---|---|---|---|
| 0.01 | $52.8 \pm 0.0$ | $0.0 \pm 0.0$ | $0.0 \pm 0.0$ | $100.0 \pm 0.0$ |
| 0.05 | $52.5 \pm 0.1$ | $7.3 \pm 0.8$ | $0.0 \pm 0.0$ | $92.7 \pm 0.3$ |
| 0.1 | $49.1 \pm 0.6$ | $48.0 \pm 0.7$ | $0.0 \pm 0.0$ | $52.0 \pm 0.2$ |
| 0.15 | $44.2 \pm 0.4$ | $68.1 \pm 0.3$ | $19.7 \pm 0.4$ | $12.2 \pm 0.1$ |
| 0.2 | $42.0 \pm 0.2$ | $77.5 \pm 0.2$ | $22.5 \pm 0.5$ | $0.0 \pm 0.0$ |
| 0.3 | $40.1 \pm 0.2$ | $98.1 \pm 0.0$ | $1.9 \pm 0.1$ | $0.0 \pm 0.0$ |
| 0.5 | $39.5 \pm 0.0$ | $100.0 \pm 0.0$ | $0.0 \pm 0.0$ | $0.0 \pm 0.0$ |

Table 3. Detailed results across different cost values $\beta_2$. Errors represent the standard deviation over multiple runs.

## G.2. MIMIC-IV Experiments

MIMIC-IV (Johnson et al., 2023) is a large collection of de-identified health-related data covering over forty thousand patients who stayed in critical care units. This dataset includes a wide variety of information, such as demographic details, vital signs, laboratory test results, medications, and procedures. For our analysis, we focus specifically on features related to *procedures*, which correspond to medical procedures performed during hospital visits, and *diagnoses* received by the patients.

Using these features, we address two predictive tasks: (1) a classification task to predict whether a patient will die during their next hospital visit based on clinical information from the current visit, and (2) a regression task to estimate the length of stay for the current hospital visit based on the same clinical information.

A key challenge in this task is the severe class imbalance, particularly in predicting mortality. To mitigate this issue, we sub-sample the negative mortality class, retaining a balanced dataset with $K = 5995$ samples, comprising 48.2% positive mortality cases and 51.8% negative mortality cases. Our model is trained on 80% of this dataset, while the remaining 20% is held out for validation. To ensure consistency in the results, we fixed the training and validation partitions.

|           | Model | $M_1$ | $M_2$ |
|-----------|-------|-------|-------|
| Accuracy  | 60.0  | 39.7  | 46.2  |
| Smooth L1 | 1.45  | 2.31  | 1.92  |

*Table 4.* Performance of the agents on the MIMIC-IV dataset, evaluated in terms of accuracy and Smooth L1 loss. We fixed the training/validation set such that the agents' knowledge remains fixed throughout the evaluation.

