# OpenReview forum: "A Two-Stage Learning-to-Defer Approach for Multi-Task Learning"
_ICML.cc/2025/Conference — ICML 2025 poster_

### Official Review · Reviewer_phu8 · 2025-03-12

**Overall Recommendation:** 3

**Summary:**

This paper presents a two-stage learning-to-defer (L2D) approach for the multi-task setting involving both classification and regression. The authors provide theoretical justification in the form of consistency bounds, as well as empirical justification in the form of validation on (multi-task) object detection and EHR analysis.

## Update after rebuttal
As stated in my rebuttal comment, I thank the authors for the clarifications and will maintain my recommendation of Weak accept.

**Claims And Evidence:**

Regarding theoretical claims, I will have to defer to domain experts to verify accuracy. Empirical claims appear sound, though it is somewhat difficult to interpret these results without being intimately familiar with L2D literature and evaluation.

**Essential References Not Discussed:**

If there are key references missing, then I am not aware of them nor qualified to propose them.

**Experimental Designs Or Analyses:**

Experimental design of empirical experiments seems appropriate – performance metrics were appropriate, and results (+ variability) were presented over 4 trials. Source code was provided in order to reproduce experimental results.

**Methods And Evaluation Criteria:**

The two validation datasets were appropriate for this unique setting.

**Other Comments Or Suggestions:**

References to tables/figures are unusual (ex: “Figure 5.1” [L372] and “Table 5.2” [L435]). These should refer to the tables/figures themselves, not the section.

**Other Strengths And Weaknesses:**

*Strengths*
- The paper is very well-written and carefully organized to logically walk the reader through each section.
- The related work is thorough, with unique contributions of this study clearly laid out relative to prior work.

*Weaknesses*
- Empirical results may be difficult to interpret for non-experts in L2D.

**Questions For Authors:**

I want to be transparent that this is outside of my area of expertise, so I may have elementary questions regarding the background of L2D and interpretation of results.

1.	When describing a medical use case of L2D, the authors write, “If the model is sufficiently confident, its diagnosis is accepted; otherwise, the decision is deferred to a medical expert who provides the final diagnosis.” I take it this is an illustrative example of the concept of deference rather than a concrete description of L2D as manifested in empirical experiments, correct? For example, in the empirical experiments, it seems that “deference” refers to accepting/rejecting (weighting) the predictions of task experts rather than deferring to, say, a human expert. Perhaps this is an obvious point to researchers like yourselves, but based on this description in the Introduction, I was expecting some sort of human-in-the-loop evaluation where low-confidence cases were deferred to an actual human expert. Is my understanding correct, and could the authors provide some way to make this distinction clearer for non-experts?
2.	In the object detection setting, why do the authors not compare performance to other approaches? Also, can the authors explain why this approach is practically useful if it never outperforms the largest model (Expert 2)?

**Relation To Broader Scientific Literature:**

The authors clearly lay out how, despite recent progress in L2D, previous two-stage L2D approaches do not address multi-task “classifier-regressor models”, which are relatively common in more complex tasks such as object detection.

**Theoretical Claims:**

I did not verify proofs due to time constraints.

---

> ### Author Rebuttal · Authors · 2025-03-29
>
> We thank the reviewer for their careful and constructive feedback. We are pleased that they found the paper well-written, the empirical setup appropriate, and the contributions clearly positioned in relation to prior work.
>
> Below, we provide clarifications on several points raised in the review.
>
> > Empirical results may be difficult to interpret for non-experts in L2D.
>
> To clarify, we conducted two distinct sets of experiments. In the object detection task, we considered three models of varying complexity (a lightweight Faster R-CNN, a medium-sized Faster R-CNN, and a large Faster R-CNN). To reflect their varying complexity and computational cost, we assigned different consultation scores to each agent: $\beta_0=0$, $\beta_1=\beta_1$, and $\beta_2=R_G\beta_1$ with $R_G$ being a ratio of GLOPs. This setup explicitly illustrates the trade-off between computational efficiency and prediction correctness, as the largest the model is, the better is its performance. Figure 1 shows that, for higher consultation costs $\beta_1$, our approach allocates queries mostly to the main lightweight model (cost $\beta_0$) or the first expert (cost $\beta_1$). Conversely, when $\beta_1$ is lower, the rejector strategically allocates queries across both experts (expert 1 and expert 2).
>
> In the EHR task, baseline approaches fail to achieve strong performance because they base allocation decisions solely on individual tasks (classification or regression separately). In contrast, our method jointly considers both tasks, resulting in more balanced and effective query allocation. We will clearly emphasize these interpretations in the revised manuscript to improve clarity.
>
> > References to tables/figures are unusual (ex: “Figure 5.1” [L372] and “Table 5.2” [L435]). These should refer to the tables/figures themselves, not the section.
>
> Thank you for pointing this out.  We have taken note and will update this in the revised manuscript.
>
> >  When describing a medical use case of L2D, the authors write, “If the model is sufficiently confident [...] I was expecting some sort of human-in-the-loop evaluation where low-confidence cases were deferred to an actual human expert. Is my understanding correct, and could the authors provide some way to make this distinction clearer for non-experts?
>
> Thank you for raising this important point. To clarify, the medical use-case mentioned in the Introduction is intended solely as an illustrative example of deference rather than a description of the concrete mechanism used in our empirical experiments. In our experiments, “deference” refers to the process of automatically selecting predictive agents (e.g models, humans) based solely on their predictions and associated costs.
>
> A key strength is its agent-agnostic nature: it does not rely on any specific internal structure, training paradigm, or decision process of the agents. As long as we have access to an agent’s predictions, our framework can learn a rejector $r\in\mathcal{R}$ that, for each input $x\in\mathcal{X}$, computes confidence scores and allocates the query to the agent deemed most reliable. Importantly, during inference, we only query the selected expert.
>
> In our empirical implementation, we have used automated black-box models or synthetic distribution as the experts. However, the framework is fully flexible and can incorporate human experts or any other decision-making system, as long as their predictions are available. We will update the manuscript to explicitly clarify this distinction, thereby reducing any potential confusion.
>
>
> > In the object detection setting, why do the authors not compare performance to other approaches? Also, can the authors explain why this approach is practically useful if it never outperforms the largest model (Expert 2)?
>
>  The primary reason we did not include comparisons with other object detection approaches is that our paper focuses explicitly on evaluating the effectiveness of the allocation mechanism within the Learning-to-Defer framework, rather than competing directly with state-of-the-art detection methods. Importantly, our method remains practically valuable even when it does not surpass the performance of the largest expert model (Expert 2). This is because Expert 2, while accurate, is typically too computationally expensive or slow to deploy on every query in realistic, resource-constrained environments. Our L2D approach provides a principled solution to this critical challenge by strategically routing queries to less computationally intensive models whenever appropriate, significantly reducing resource usage without severely impacting accuracy. Furthermore, when expert models have specialized strengths or operate on slightly different distributions—as demonstrated clearly in our EHR experiments—our method can exploit these discrepancies to effectively improve overall system performance beyond what any single agent could achieve alone.

---

> > ### Comment · Reviewer_phu8 · 2025-04-08
> >
> > I appreciate the authors' clarifications and will likely maintain my recommendation of Weak accept.

---

### Official Review · Reviewer_oXmZ · 2025-03-12

**Overall Recommendation:** 4

**Summary:**

The paper presents a new application of learning to defer to the context of multitask, where the task's target consists of both a regression and a classification task. The authors provide a theoretical analysis of two-stage L2D, showing that the proposed surrogate loss is both Bayes-consistent and $\mathcal{G}, \mathcal{R}$ consistent.
Empirical results showcase the effectiveness of the approach.

**Claims And Evidence:**

I think the paper is convincing enough in terms of contribution and impact of the approach in real-life scenarios.
The theoretical analysis correctly supports the paper's claims.

**Essential References Not Discussed:**

The essential literature is correctly discussed, the only reference I think must be necessarily added is the work by Okati et al., 2021, which considers a formulation with explicit constraints in terms of coverage.

[Okati et al., 2021] - Okati, N., De, A., & Rodriguez, M. (2021). Differentiable learning under triage. NeurIPS 2021

**Experimental Designs Or Analyses:**

The analyses provided are sufficiently sound, with enough details to reproduce results. Overall, the main concern I have is regarding the choice of subsampling the dataset for MIMIC IV, as detailed in Appendix G.2. Other options could be available, e.g., weighting the loss between classes differently, and the authors could have commented a bit more this aspect.

**Methods And Evaluation Criteria:**

The proposed datasets and evaluation criteria make sense. The authors consider two datasets, one that is new in the setting of learning to defer, and one used also in previous works [Mao et al., 2023a, 2024e]. Hence, the empirical evaluation makes sense overall.

**Other Comments Or Suggestions:**

I provide here for completeness a few recent articles, which could enlarge the current related work section (as they expand different aspects of Learning to Defer):

- In [Wei et al., 2024], the authors provide a refinement of Bayes consistency, called Bayes-dependent consistency;
- In [Palomba et al., 2025], the authors bridge causal inference and learning to defer for improved evaluation of such systems;
- In [Strong et al., 2025], the authors present an application of L2D for healthcare using LLMs

Finally, according to ICML guidelines, the caption for tables should be placed above. Please consider adjusting the ones in the paper.


[Wei et al., 2024] - Wei, Z., Cao, Y., & Feng, L. (2024). Exploiting human-AI dependence for learning to defer. ICML '24

[Palomba et al., 2025] - Palomba, F., Pugnana, A., Alvarez, J., Ruggieri, S. (2025). A Causal Framework for Evaluating Deferring Systems. AISTATS '25

[Strong et al., 2025] - Strong, J., Men, Q., & Noble, A. (2025). Towards Human-AI Collaboration in Healthcare: Guided Deferral Systems with Large Language Models. AAAI '25

**Other Strengths And Weaknesses:**

The paper is clearly written, with convincing motivations for the proposed approach.

**Questions For Authors:**

I think the paper is sound enough, with good motivation and adequate evaluation.
Hence, I am prone to suggesting acceptance of the paper.

I have a couple of questions for the authors:

- Have the authors considered how to directly model the coverage for experts, as considered for instance in [Okati et al.,2021; Mozannar et al., 2023; Palomba et al., 2025]?
- Could the authors comment on my concern for the choice of subsampling the dataset? I guess adding a weighted loss is a safe option, but I do not want to overlook consistency concerns.

### Update After Rebuttal

I am satisfied with the answers from the authors. I keep my positive score.

**Relation To Broader Scientific Literature:**

The paper is correctly positioned in the literature, with most works in learning to defer correctly referenced.
Moreover, the work extends existing literature,  considering cases where regression and classification occur simultaneously. I think this is a valuable contribution with sound proof and analysis.

**Theoretical Claims:**

The proofs seem correct, but I might have missed some details.

---

> ### Author Rebuttal · Authors · 2025-03-29
>
> We sincerely thank the reviewer for their constructive feedback. We appreciate their positive assessment of the strength of our theoretical and empirical contributions.
>
> Below, we address the potential connection to coverage constraints and provide further details on specific aspects of our experimental design.
>
> > The analyses provided are sufficiently sound, with enough details to reproduce results. Overall, the main concern I have is regarding the choice of subsampling the dataset for MIMIC IV, as detailed in Appendix G.2. Other options could be available, e.g., weighting the loss between classes differently, and the authors could have commented a bit more this aspect.
>
> > Could the authors comment on my concern for the choice of subsampling the dataset? I guess adding a weighted loss is a safe option, but I do not want to overlook consistency.
>
> Thank you for raising this point. Indeed, alternative heuristics such as class-weighted losses could have been employed. However, our primary objective was not necessarily to optimize overall performance metrics, but rather to highlight the relative improvement and effectiveness of our approach. Nevertheless, your suggestion of using weighted losses is valid and aligns with our theoretical framework, as our consistency proofs hold for any positive cost $c_j$. A promising direction for future experiments would involve adapting agent-specific cost functions in a cost-sensitive manner to address class imbalance explicitly.
>
> > The essential literature is correctly discussed, the only reference I think must be necessarily added is the work by Okati et al., 2021, which considers a formulation with explicit constraints in terms of coverage.
>
> This is correct, we will add this paper in the related work.
>
> > Have the authors considered how to directly model the coverage for experts, as considered for instance in [Okati et al.,2021; Mozannar et al., 2023; Palomba et al., 2025]?
>
> This is an excellent question. In the single-expert setting considered in [5,6,7], modeling coverage explicitly is relatively straightforward. Consider a rejector $r:\mathcal{X}\rightarrow\mathbb{R}$, where $r(x)\geq 0$ indicates deferring to the expert, and $r(x)<0$ indicates no deferral. To directly model coverage, one can introduce a coverage level $k \in \mathbb{R}^+$ corresponding to the $\overline{k}^{\,\text{th}}$ percentile of the rejector scores $r(x)$ computed over a validation set. This yields an adjusted decision rule: $1\_{r(x)\geq k}$, instead of $1\_{r(x)\geq 0}$, resulting in an expected coverage of $\mathbb{E}[1\_{r(x)\geq k}] = 1-\overline{k}$.
>
> However, in the multi-expert setting, extending this idea is nontrivial because deferral decisions involve complex allocations across multiple experts. This complexity likely explains why [5,6,7] primarily focus on single-expert scenarios. **One possible direction to handle coverage explicitly in multi-expert settings is to consider margin-based rejectors**. Let $\mathcal{A}= \lbrace 0,\dots,J \rbrace$ denote the set of $J$ experts plus a main model, and define a rejector $r:\mathcal{X}\rightarrow\mathcal{A}$. We then define the margin-based score as
>
> $$\rho\_r(x,j)=r(x,j)-\max\_{j'\neq j} r(x,j'),$$
>
> where a positive margin $\rho\_r(x,j)\geq 0$ leads to deferring to expert $j$. Coverage can be incorporated by setting thresholds $\tilde{k}\_j$ separately for each expert $j$, corresponding to specific percentiles of the margin $r(x,j)-r(x,0)$. The decision rule then becomes $1\_{\rho\_r(x,j)\geq \tilde{k}\_j}$. **Importantly, this approach naturally reduces to the single-expert coverage** definition used in [5,6,7], since deferral occurs when $\rho(x,1)\geq 0$, i.e., $r(x,1)\geq r(x,0)+\tilde{k}\_1$.
>
> We hope this aligns with your question and would be glad to discuss it further. We will include this clarification and an expanded discussion in the revised manuscript.
>
> ### References
> [1] Mao, et al. (2023). Two-Stage Learning to Defer with Multiple Experts. NeurIPS23
>
> [2] Mao, et al. (2024). Regression with multi-expert deferral. NeurIPS24
>
> [3] Narasimhan et al. (2022) Post-hoc estimators for learning to defer to an expert. NeurIPS22
>
> [4] Mao et al. (2024) Theoretically grounded loss functions and algorithms for score-based multi-class abstention. AISTATS24
>
> [5] Okati et al. (2021). Differentiable learning under triage. NeurIPS21
>
> [6] Mozannar et al. (2023, April). Who should predict? exact algorithms for learning to defer to humans. AISTATS23
>
> [7] Palomba et al. (2025). A Causal Framework for Evaluating Deferring Systems. AISTATS25

---

> > ### Comment · Reviewer_oXmZ · 2025-04-02
> >
> > I thank the authors for their rebuttal and for addressing my questions. I am satisfied with their answers. Overall, I think this is a valuable contribution. Thus, I am keeping my acceptance score.

---

### Official Review · Reviewer_cfAF · 2025-03-14

**Overall Recommendation:** 2

**Summary:**

The paper developed a Two-Stage Learning-to-Defer framework for multi-task problems, enabling joint classification and regression. This framework features a novel two-stage surrogate loss family that is both $(\mathcal{G}, \mathcal{R})$-consistent and Bayes-consistent for cross-entropy-based surrogates. The authors derived tight consistency bounds and established minimizability gaps, extending prior work on Learning-to-Defer. Their learning bounds improve with richer hypothesis spaces and more confident experts. The authors validated the approach on object detection and electronic health record analysis, demonstrating its superiority over existing methods.

**Claims And Evidence:**

Yes.

**Essential References Not Discussed:**

N/A.

**Experimental Designs Or Analyses:**

Yes.

**Methods And Evaluation Criteria:**

Yes.

**Other Comments Or Suggestions:**

N/A.

**Other Strengths And Weaknesses:**

Strengths: The paper is well-written, clearly presented, and provides a detailed discussion of prior work, making it accessible to non-experts.

Weaknesses: The results, while sound, seem to offer incremental progress rather than significant novelty compared to existing work. Furthermore, I question whether classification can be viewed as a specific instance of regression, with the zero-one loss as the evaluation metric. If so, would the proposed framework be adequately addressed by existing regression-based deferral frameworks?

**Questions For Authors:**

1. Is it possible to view classification as a specific instance of regression, where the zero-one loss serves as the evaluation metric? If this perspective is valid, would the current framework be adequately addressed by the regression-based deferral framework presented in previous research?

2. What specific challenges arise when adapting the learning-to-defer framework for multi-task learning scenarios?

**Relation To Broader Scientific Literature:**

Section 2 provides a sound discussion.

**Theoretical Claims:**

Yes.

---

> ### Author Rebuttal · Authors · 2025-03-29
>
> We thank the reviewer for their careful reading and thoughtful comments. We are pleased that they found our framework well-presented and the theoretical analysis rigorous.  In the following, we clarify the technical challenges that arise in the multi-task setting.
>
> First, we emphasize that **our paper provides theoretical novelty**. Our framework is generalized to accommodate any positive cost $c\_j$, enabling a unified treatment of classification, regression, and multi-task problems within Learning-to-Defer. We derived the Bayes-optimal rejector and established consistency guarantees for multi-task L2D. In particular, we prove consistency bounds that hold for any surrogate in the comp-sum family—parameterized by $\nu$ as defined in Equation 1—which encompasses commonly used surrogates such as the logistic, MAE, and exponential losses. These bounds are both tighter and more interpretable than those in [1, 2, 3], accounting for both the parameter $\nu$ and the $L\_1$ norm of the aggregated cost vector $\boldsymbol{\tau}$ (see discussion with reviewer @oatj). Additionally, we present a novel analysis of minimizability gaps in Theorem 4.6, demonstrating that the optimal conditional risk critically depends on both the norm of the aggregated costs $\boldsymbol{\tau}$ and the choice of the multiclass surrogate $\Phi\_{01}^\nu$ (Equation 1). Notably, our results hold without any assumptions on the underlying distribution $\mathcal{D}$—in contrast to [4]—and apply to any surrogate $\Phi\_{01}^\nu$ within the comp-sum family. Furthermore, we provide new generalization bounds specifically tailored to this setting.
>
> While classification can conceptually be viewed as a special case of regression under the zero-one loss, such a simplification overlooks important theoretical distinctions that arise when analyzing consistency and optimality in deferral decisions. To highlight this, we give some additional analysis:
>
> Let $g(x) = (h \circ w(x), f \circ w(x))$ be any multi‐head model where $w \in \mathcal{W}$ is a shared representation and $h \in \mathcal{H}$, $f \in \mathcal{F}$ are heads for classification and regression, respectively. Define $L(w,h,f) = \mathbb{E}\_{y,t \mid x}[c\_0(h(w(x)),f(w(x)),z)]$ as the conditional risk for the triple $(w,h,f)$. Let $L^\ast = \inf\_{(w,h,f)} L(w,h,f)$ be the optimal multi‐head conditional risk in the hypothesis class $\mathcal{G} = \lbrace x \mapsto (h(w(x)), f(w(x))) \mid w \in \mathcal{W}, h \in \mathcal{H}, f \in \mathcal{F} \rbrace$. We then write $L(w,h,f) - L^\ast$ to denote the conditional excess risk of $(w,h,f)$ above the best multi‐head. Observe that for a chosen triple $(w,h,f)$, we can decompose:
> $$L(w,h,f) - L^\ast = \bigl[L(w,h,f) - \inf\_{h',f'} L(w,h',f')\bigr] + \bigl[\inf\_{h',f'} L(w,h',f') - L^\ast\bigr],$$
>
> with $h'$ and $f'$ acting directly on $x$. Define the heads gap $\Delta\_{\mathrm{heads}}(w,h,f) = L(w,h,f) - \inf\_{h',f'} L(w,h',f')$, which measures how well the specific heads $(h,f)$ perform given a fixed representation $w$. Then define the representation gap $\Delta\_{\mathrm{repr}}(w) = \inf\_{h',f'} L(w,h',f') - L^\ast$, which captures how close $w$ is to the best possible representation for both tasks. From this notation, we rewrite
>
> $$L(w,h,f) - L^\ast = \Delta\_{\mathrm{heads}}(w,h,f) + \Delta\_{\mathrm{repr}}(w).$$
>
> We next introduce a multi‐task comparison by defining a separate‐training conditional risk. Let $L\_{\mathrm{sep}}(h',f') = \mathbb{E}\_{y,t \mid x}[c\_0((h'(x),f'(x)),z)]$ be the conditional risk of using completely separate models $h'$ and $f'$ without any shared representation. Let $L\_{\mathrm{sep}}^\ast = \inf\_{h',f'} L\_{\mathrm{sep}}(h',f')$. We measure the quality of forcing a single shared $w$ via
>
> $$\Delta\_{\mathrm{MTL}} = L^\ast - L\_{\mathrm{sep}}^\ast.$$
>
> If $\Delta\_{\mathrm{MTL}} < 0$, then joint training (i.e. a shared $w$) is strictly better than separate solutions; if $\Delta\_{\mathrm{MTL}} > 0$, it is worse. Finally, we can link $\Delta\_{\mathrm{MTL}}$ to the multi‐head conditional excess risk by noting
>
> $$L(w,h,f) - L^\ast = \bigl[L(w,h,f) - L\_{\mathrm{sep}}^\ast\bigr] - \Delta\_{\mathrm{MTL}}.$$
>
> Hence, if $\Delta\_{\mathrm{MTL}} < 0$, it follows that $L(w,h,f) - L^\ast < L(w,h,f) - L\_{\mathrm{sep}}^\ast$, meaning the multi‐head optimum is easier to approach than the separate‐training optimum. Conversely, if $\Delta\_{\mathrm{MTL}} > 0$, we get $L(w,h,f) - L^\ast > L(w,h,f) - L\_{\mathrm{sep}}^\ast$, so it is harder to reach the best multi‐head risk than to reach the best separate‐training solution.
>
> Following the discussion with Reviewer @oXmZ, we will also incorporate a discussion on how our approach can be adapted to account for coverage constraints.
>
> We hope this clarifies your question. We will add those novel analysis and revise the manuscript to make this point clearer in the final version.
>
> Please refer to the discussion with reviewer @oXmZ for references.

---

### Official Review · Reviewer_oatj · 2025-03-14

**Overall Recommendation:** 3

**Summary:**

In this paper, the authors analyze the learning-to-differ (L2D) problem in the two-staged multi-task (classification and regression) setting. The paper introduces the pointwise Bayes rejector for the mult-task deferral and introduces a surrogate differal loss that is Bayes consistent. The paper further provides generalization guarantees for the learned rejector, which provide insights into the conditions of the problem setting that can lead to better generalization. Numerical experiments are provided to validate the proposed method, and a comparison of the proposed multi-task rejector with prior single-task rejectors shows that the proposed method can have a balanced trade-off between the tasks compared to prior methods.

**Claims And Evidence:**

Yes.

**Essential References Not Discussed:**

None.

**Experimental Designs Or Analyses:**

* It is unclear how the coefficients $\lambda^\text{cla}, \lambda^\text{reg}$ used in the agents' cost were determined. These coefficients seem crucial in attaining the balanced performance described in the discussion.
* It is unclear how the rejection is implemented for the prior methods in the experiments in Section 5.2 in the multi-task setting. For example, when the classification rejector is triggered, does the model consult for the expertise on both classification and regression tasks from the expert, or only the classification task?

**Methods And Evaluation Criteria:**

Yes.

**Other Comments Or Suggestions:**

* It seems like in proof of Lemma 4.2, $C^B_{\ell_{\text{def}}}$ should not depend on $g, r$.
* $\bar{\tau}$ is not defined in Theorem 4.4, and it is not clear where $T^{-1, \nu}, T^{\nu=1}$ appear in the Theorem.
* The clusters $C_{cla}^{M1}, C_{cla}^{M2}$ defined in Experiments in 5.2 seem not distinct as described.

**Other Strengths And Weaknesses:**

Strengths

* The paper provides theoretical insights into L2D in the two-staged multi-task setting, which seems novel in the related literature.
* The paper is mostly self-contained and easy to follow
* The problem is well-motivated, and the theoretical insights are sufficiently discussed.

Weaknesses

* The definition of the deferral function class $L_{def}$ (column 1 line 319) to be a mapping to [0,1] seems unrealistic, given that the $\ell_{def}$ is made up of $c_0$, which in turn is assumed to be a summation of $\ell_{01}$ and $\ell_{reg}$. Since $\ell_{reg}$ is not necessarily bounded between 0 and 1, it is unclear why the definition of $L_{def}$ is correct.
* Given the possibly different scales of classification and regression losses, more discussion seems to be needed regarding designing a balanced/meaningful rejector in this setting.

**Questions For Authors:**

* Please refer to the questions/concerns raised in previous sections.

**Relation To Broader Scientific Literature:**

This paper extends the literature in the area of L2D, by providing theoretical insights into L2D in the two-staged multi-task setting.

**Theoretical Claims:**

Did not check all the proofs, minor typos in the proof of Lemma 4.2 (mentioned in Other Comments Or Suggestions).

---

> ### Author Rebuttal · Authors · 2025-03-29
>
> We thank the reviewer for their detailed and thoughtful feedback. We are glad that they found our theoretical contributions novel, the paper well-motivated, and the analysis of the two-stage multi-task Learning-to-Defer setting valuable to the literature. Please find some clarification below:
>
> > It is unclear how the coefficients $\lambda^{cla}$ and $\lambda^{reg}$ [...] balanced performance described in the discussion.
>
> > Given the possibly different scales of classification and regression losses, more discussion seems to be needed regarding designing a balanced/meaningful rejector in this setting.
>
> We agree that the choice of the coefficients $\lambda^{cla}$ and $\lambda^{reg}$ is crucial for achieving balanced and meaningful performance. In our experiments, we considered both classification and regression tasks equally important; thus, we set $\lambda^{cla} = \lambda^{reg} = 1$. This choice represents a balanced and task-agnostic baseline, ensuring no implicit bias toward either component. In practical scenarios, these coefficients should indeed be tuned according to task-specific priorities and performance requirements. **For instance, if the classification is more important in this setting, we should set $\lambda^{cla}>\lambda^{reg}$ to prioritize agent with more proficiency on the classification task**. We will explicitly discuss this and highlight the importance of tuning $\lambda^{cla}$ and $\lambda^{reg}$ in the revised manuscript.
>
> > It is unclear how the rejection is implemented for the prior methods [...] or only the classification task?
>
> In the EHR task, we treat classification and regression as independent but simultaneously allocated tasks. Specifically, when the classification rejector (as described in [1]) is triggered, we allocate both the classification and regression queries jointly to the selected agent. Similarly, when the regression rejector is triggered, it also allocates both tasks together [2]. We will clarify this joint-allocation implementation explicitly in the revised manuscript.
>
> >The definition of the deferral function class (column 1 line 319) to be a mapping to $[0,1]$ seems unrealistic.
>
> > It seems like in proof of Lemma 4.2, $\mathcal{C}\_{\ell\_{def}}^B$ should not depend on $g,r$.
>
> Thank you for pointing this out. You are correct—these are typos and do not affect the correctness of the proofs. We will fix them in the revised manuscript.
>
> > $\overline{\tau}$ is not defined in Theorem 4.4, and it is not clear where $T^{1-\nu}$ appear in the Theorem.
>
> Indeed, $\overline{\tau}\_j$ represents the expected aggregated cost of agent $j \in \mathcal{A}$, formally defined as $\overline{\tau}\_j = \mathbb{E}_\{y,t|x}[\tau\_j]$. Consequently, we define the vector $\overline{\boldsymbol{\tau}} = (\overline{\tau}\_0, \overline{\tau}\_1, \dots, \overline{\tau}\_J)$ and $||\boldsymbol{\overline{\tau}}||\_1$ its L1 norm.
>
> The transformation function $\mathcal{T}^{\nu}(u)$ is introduced to determine the function $\Gamma^{\nu}(u)$ corresponding to a given multiclass surrogate loss $\Phi^\nu\_{01}$. **We explicitly show that the consistency bounds of the deferral loss (Lemma 4.3) inherently depends on the consistency bounds of the multiclass surrogate loss** $\Phi^\nu_{01}$ from the comp-sum family (Equation 1). For example, choosing the log-softmax surrogate corresponds to setting $\nu = 1$. In this case, the relevant transformation function is explicitly given by $\mathcal{T}^1(u)=\frac{1+u}{2} \log[1+u] + \frac{1-u}{2} \log[1-u]$ [8]. Then, by taking the inverse of this transformation, we obtain the function $\Gamma^\nu(u)=(\mathcal{T}^\nu)^{-1}(u)$ leading to a bound with the function  $\overline{\Gamma}^\nu(u)=||\overline{\boldsymbol{\tau}}||\_1 \Gamma^\nu(\frac{u}{||\overline{\boldsymbol{\tau}}||\_1})$ depending on both the surrogate transformation $\mathcal{T}^{\nu}(u)$ and the L1 norm $||\overline{\boldsymbol{\tau}}||\_1$.
>
> We will explicitly include these definitions and clarifications in the revised manuscript.
>
> > The clusters $C_{cla}^{M_1}, C_{cla}^{M_2}$ defined in Experiments in 5.2 seem not distinct as described.
>
> We confirm that clusters $C\_{cla}^{M_1}$ and $C\_{cla}^{M_2}$ were explicitly constructed to represent distinct regions of expertise, although they do indeed exhibit some overlap. Upon review, we acknowledge a minor inconsistency in the original manuscript regarding their definitions (but not on experiments). The correct cluster assignments are $C^{M\_1}\_{cla} = \lbrace C\_1, C\_2, C\_4\rbrace$ and $C^{M\_2}\_{cla} = \lbrace C\_1, C\_5, C\_6 \rbrace$. We will correct this in the revised manuscript.
>
> ### References
> [1] Mao, et al. (2023). Two-Stage Learning to Defer with Multiple Experts. NeurIPS23
>
> [2] Mao, et al. (2024). Regression with multi-expert deferral. NeurIPS24
>
> [8] Mao et al. (2023). Cross-entropy loss functions: theoretical analysis and applications. ICML23

---

### Decision · Program_Chairs · 2025-05-01

**Decision:**

Accept (poster)

**Comment:**

This paper examines two-stage learning to defer under the multi-task setting, meaning that there are supervised losses for both classification and regression, providing a theoretical analysis of the consistency of the proposed loss as well as empirical validation on the Pascal VOC and MIMIC datasets.  The reviewers generally had positive opinions of the work (1 x accept, 2 x weak accept, 1 x weak reject).   The reviewer who leaned toward rejection's primary criticism is that the extension to multi-task learning had marginal novelty.  The authors provided a detailed rebuttal, nicely explaining the paper's theoretical contributions, which I found convincing.

Small comment: I noticed two errors in the years of the citations---one was that Verma et al. should be AIStats 2023, not 2022.  I can't remember the other one